# BANDITQ : Fair Bandits with Guaranteed Rewards

**Abhishek Sinha**[1]

[1] School of Technology and Computer Science, Tata Institute of Fundamental Research, Mumbai 400005, India
abhishek.sinha@tifr.res.in

## Abstract

Classic no-regret multi-armed bandit algorithms, including the Upper Confidence Bound (UCB), HEDGE, and EXP3, are inherently unfair by design. Their unfairness stems from their very objective of playing the most rewarding arm as frequently as possible while ignoring the rest. In this paper, we consider a fair prediction problem in the stochastic setting with a guaranteed minimum rate of accrual of rewards for each arm. We study the problem in both full-information and bandit feedback settings. Combining queueing-theoretic techniques with adversarial bandits, we propose a new online policy called BANDITQ that achieves the target reward rates while conceding a regret and target rate violation penalty of at most $O(T^{3/4})$. The regret bound in the full-information setting can be further improved to $O(\sqrt{T})$ under either a monotonicity assumption or when considering time-averaged regret. The proposed policy is efficient and admits a black-box reduction from the fair prediction problem to the standard adversarial MAB problem. The analysis of the BANDITQ policy involves a new self-bounding inequality, which might be of independent interest.

## 1 INTRODUCTION

A vast majority of the multi-armed bandit (MAB) algorithms deployed in practice are designed to maximize the cumulative rewards. Consequently, these algorithms could end up systematically avoiding a subset of arms (which could represent users with certain demographic characteristics or historical activities) that the algorithm finds less rewarding (Sweeney, 2013). In a typical case of algorithmic discrimination, Facebook was sued for targeting ads on housing, credit and employment based on race, gender, and religion -

all protected classes under US law (Hao, 2019). A similar problem of fair allocation of resources arises in wireless settings, where schedulers maximizing the total throughput could result in not serving a subset of users having relatively poor channels. A number of papers have proposed a solution to the fairness problem by putting an explicit constraint on the *minimum frequency* of pulls for each arm. However, in many problems of practical interest, the algorithm designer is interested in guaranteeing a minimum rate of *reward accrual* for each arm - not just ensuring a minimum frequency at which the arms ought to be pulled.

**Examples:** (1) In online ad allocation, the advertisers are primarily interested in maximizing their click-through rate, which fetches them monetary rewards, rather than just the number of times their ads are displayed against a search result. (2) In wireless scheduling problems, the users, who correspond to the bandit's arms in our formulation, are interested in guaranteed data rates rather than their frequency of scheduling - a low-level metric transparent to the users. (3) As a final example, consider a crowdsourcing platform (e.g., Amazon Mechanical Turk) where the workers receive payments for performing tasks (Fu and Liu, 2021). Upon completing each task, the platform receives a fixed percentage of the payment as revenue. The goals of the platform are - (a) to allocate the oncoming tasks fairly among the workers and (b) to maximize the platform's total revenue. In our formulation, the workers correspond to the arms, and the revenue maximization problem (b) becomes equivalent to the regret minimization problem. However, without the fairness requirement (a), the platform would assign most of the jobs to the best-performing workers, effectively ignoring a vast majority of the registered workers who may leave the platform dissatisfied. Hence, the platform may suffer from a high attrition rate. One possible way to enhance the retention rate of the workers and make the platform non-discriminating is to ensure a guaranteed reward rate (equivalent to a minimum wage) for each registered worker. In this paper, we will see that the proposed BANDITQ policy gives an efficient solution to each of the above problems.

Clearly, the rate of reward accruals of the arms depends on the unknown reward distribution, which needs to be learned along the way. In this paper, we solve this fair prediction problem in the stochastic setting via a black-box reduction to an adversarial MAB problem by making use of a natural queueing dynamics to keep track of the target rates. Although we consider i.i.d. rewards, we will see that the use of adversarial MAB sub-routines is essential to account for the target reward rate constraints.

## 1.1 RELATED WORKS

There is extensive literature on the classic Multi-armed Bandits (MAB) problem, where the objective is to sequentially play an arm on each round from a given set of arms with unknown reward distributions to maximize the cumulative reward. As the feedback is limited to the observed rewards only, the MAB problem naturally involves an exploration vs exploitation trade-off. See Cesa-Bianchi and Lugosi (2006); Bubeck et al. (2012); Lattimore and Szepesvári (2020) for textbook treatments on MAB. The fair prediction problem considered in this paper belongs to a class of MAB problems with global constraints. Several authors have considered variants of the fair prediction problem in MAB with widely varying definitions for fairness (Joseph et al., 2016; Gillen et al., 2018; Bechavod et al., 2020; Hossain et al., 2021; Huang et al., 2022). Closer to our setting, the papers by Patil et al. (2021); Claure et al. (2020), and Li et al. (2019) considered a stochastic MAB problem while requiring the minimum *fraction* of pulls of each arm to exceed a given threshold. Celis et al. (2019) considered a similar problem in the personalized recommendation setting where both the minimum and the maximum fraction of pulls are constrained in order to avoid the polarization of views. Similar to ours, Li et al. (2019) used a virtual queueing recursion to handle the fairness constraints. However, their UCB-based policy yields a regret bound which varies *linearly* with the horizon length (Li et al., 2019, Theorem 2). Chen et al. (2020) considered the above problem in the contextual bandit setting and proposed a no-regret policy with a known context distribution. Cai et al. (2018) considered a related stochastic MAB problem with a long-term constraint on an auxiliary (level-2) reward process, which is assumed to be *independent* of the main (level-1) rewards of the arms. On the other hand, in our problem, the corresponding level-1 and level-2 reward processes are identical, and hence, these results do not apply due to the lack of the independence assumption. Badanidiyuru et al. (2018); Immorlica et al. (2022), and Xia et al. (2015) considered the Bandits with Knapsack (BwK) problem in the stochastic and adversarial settings. In this problem, a given resource budget is allocated to the arms at the beginning, and the policy continues until one of the arms finishes all of its budgets. Immorlica et al. (2022) used a Lagrangian-based technique to design a no-regret policy for the BwK problem. A recent paper by Bistritz and Bambos

(2022) considered a similar multiplayer multi-armed bandit problem with QoS constraints. However, they did not provide any regret bound. In this connection, we also mention a parallel line of work on fair resource allocation policies where, instead of meeting explicit constraints, the objective is to maximize a non-linear concave utility function of the cumulative rewards (Sinha et al., 2023). Our problem is also closely connected to a recent series of works on Online Convex Optimization (OCO) with long-term constraints (Neely and Yu, 2017; Yu et al., 2017; Yuan and Lamperski, 2018; Castiglioni et al., 2022). While these papers propose problem-specific policies, we give a black-box reduction using any arbitrary adaptive learning policy as a subroutine and achieve state-of-the-art regret and constraint violation bounds. Furthermore, while most of the previous papers consider the full-information setting and/or assume the strict feasibility or Slater's condition, we consider the more general bandit feedback setting *without* making any additional assumptions. The Lyapunov-based technique presented in this paper has been recently extended to solve the problem of OCO with long-term constraints as well (Sinha and Vaze, 2023, 2024).

## 1.2 OUR CONTRIBUTIONS

In contrast with a major line of work on fair MABs, which is mainly concerned with guaranteeing a minimum frequency of plays for each arm (*procedural fairness*), in this paper, we initiate the study of a class of problems guaranteeing a minimum *rate* of reward accruals for each arm (*substantive fairness*). Compared to the standard MAB problem, here, the difficulty stems from the fact that in addition to playing the unknown best arm sufficiently many times, other arms with unknown mean rewards also ought to be played frequently enough so as to satisfy the given fairness constraints. Consequently, the design of our algorithm and its analysis proceed along a different line from that of the prior works. In particular, we claim the following contributions:

1. We propose a fair learning policy for stochastic bandits, called BANDITQ, via a *black-box* reduction to the standard adversarial MAB problem. The problem is studied in both full information and bandit feedback settings. The proposed BANDITQ policy keeps track of the global reward rate constraints through an auxiliary queueing process, which is then used to define the rewards for the unconstrained MAB problem recursively.

2. An attractive feature of our policy is that it is completely oblivious to the algorithm used for the unconstrained MAB problem. In particular, BANDITQ can use *any* existing MAB policy with a data-dependent adaptive regret bound. The key to this attractive separation result is a new *self-bounding* inequality that bounds the sum of the regret and current rate violations in terms of past violations.

3. We introduce a new proof technique that bounds the regret and rate violations by solving certain sequential inequalities. The proof arguments are crisp and utilize off-the-shelf adaptive regret bounds.

4. We supplement our theoretical results with illustrative numerical experiments.

## 2 PROBLEM FORMULATION

We consider a regret minimization problem in the context of Multi-armed Bandits (MAB) with an additional fairness constraint. The fairness constraint requires that each arm in a given subset $\mathcal{P}$ (called *protected class*) must attain pre-specified reward accrual rates, which are assumed to be feasible. Formally, we consider an $N$-armed bandit, which on round $t$ receives an unknown reward vector $\boldsymbol{r}(t) \in [0,1]^N$. The vector $\boldsymbol{r}(t)$ is generated i.i.d. on each round with an unknown expectation $\boldsymbol{\mu}$. On round $t$, an online policy first decides a probability distribution $\boldsymbol{x}(t) \in \Delta_N$, where $\Delta_N$ denotes the set of all probability distributions supported on $N$ arms. The policy then randomly samples an arm $I_t \in [N]$ from the distribution $\boldsymbol{x}(t)$[1]. Depending on the feedback structure, either the entire reward vector $\boldsymbol{r}(t)$ (in the case of full information feedback) or the reward of the sampled arm $r_{I_t}(t)$ only (in the case of bandit feedback) is revealed to the policy at the end of round $t$. The above process continues for $T$ rounds.

**Fairness constraints:** Due to the action of the policy, the selected arm $I_t$ receives a random reward of value $r_{I_t}(t)$. Hence, if on round $t$, the online policy samples arms according to the distribution $\boldsymbol{x}(t)$, the $i^{\text{th}}$ arm receives a (conditional) expected reward of $x_i(t)\mathbb{E}r_i(t) = x_i(t)\mu_i$, and the online policy receives an overall (conditional) expected reward of $\langle \boldsymbol{x}(t), \boldsymbol{\mu} \rangle$. Let $\vec{\lambda}$ be the given target reward rates vector. The fairness constraint mandates that the long-term rate of rewards accrued by the arm $i \in \mathcal{P}$ must be at least $\lambda_i, \forall i \in \mathcal{P}$ (see Eqn. (4)). For notational simplicity, we may assume that $\lambda_i = 0, \forall i \in [N] \setminus \mathcal{P}$.

**Offline Benchmark and Performance Metric:** We compare the performance of an online policy against any fixed sampling distribution $\boldsymbol{x}^* \in \Delta_N$ that meets the target reward rates. In other words, our comparator class $\Omega(\vec{\lambda})$, indexed by the target vector $\vec{\lambda}$, is defined as follows:

$$\Omega(\vec{\lambda}) = \left\{ \boldsymbol{x}^* \subseteq \Delta_N : x_i^* \mu_i \geq \lambda_i, \ \forall i \in \mathcal{P} \right\}. \quad (1)$$

Clearly, in order for the target rate vector $\vec{\lambda}$ to be feasible (*i.e.,* $\Omega(\vec{\lambda}) \neq \varnothing$), it is necessary and sufficient that

$$\sum_i \frac{\lambda_i}{\mu_i} \leq 1. \quad (2)$$

See Section A in the Appendix for a brief discussion on the feasibility assumption. The set of all offline benchmarks $\Omega(\vec{\lambda})$ is closed and convex with an Euclidean diameter of $D = \sqrt{2}$. Our goal is to design a sampling policy $\{\boldsymbol{x}(t)\}_{t \geq 1}$ that achieves a sublinear regret against any $\boldsymbol{x}^* \in \Omega(\vec{\lambda})$, where

$$\text{Regret}_T(\boldsymbol{x}^*) \equiv \langle \boldsymbol{x}^*, \boldsymbol{\mu} \rangle T - \mathbb{E} \sum_{t=1}^{T} \sum_{i=1}^{N} r_i(t) \mathbb{1}(I_t = i), \quad (3)$$

while meeting the long-term reward rate constraints defined next[2]. Asymptotically, for any time interval $\mathcal{I} \subseteq [T]$, the long-term rate constraint requires:

$$\liminf_{|\mathcal{I}| \to \infty} |\mathcal{I}|^{-1} \mathbb{E}\Big[ \sum_{t \in \mathcal{I}} r_i(t) \mathbb{1}(I_t = i) \Big] \geq \lambda_i, \ \forall i \in \mathcal{P}. \quad (4)$$

Note that Eq. (4) requires the minimum reward rate guarantee to hold *uniformly* across the time horizon for any sufficiently long interval of time. In other words, we require that no individual arm is starved for a long period of time - a problem left open by Patil et al. (2021). Furthermore, following Cai et al. (2018), we work with a fine-grained non-asymptotic metric *rate violation penalty* defined below:

$$\mathbb{V}(T) = \max_{i \in P} \mathbb{E}\Big[ \sum_{t=1}^{T} \big( \lambda_i - r_i(t) \mathbb{1}(I_t = i) \big) \Big]. \quad (5)$$

In brief, we seek to design an online policy for which *both* $\text{Regret}_T$ and $\mathbb{V}(T)$ are sub-linear in $T$. We note two fundamental differences between the above problem and the standard online learning framework (Orabona, 2019). First, contrary to the online learning setting, where the set of benchmarks $\Omega$ is specified a priori (independent of the rewards), in this problem, the set of benchmarks (1) depends on the unknown reward distributions through their expectations $\boldsymbol{\mu}$. Second, unlike the online learning setting, the action taken by the policy on a round is not restricted to the set $\Omega(\vec{\lambda})$ provided that the long-term target rates are met. Note that upon setting the vector $\vec{\lambda}$ to zero, we recover the classic MAB problem as a special case. In the following section, we introduce the BANDITQ policy in the full information setting.

## 3 BANDITQ POLICY WITH FULL INFORMATION FEEDBACK

In this Section, we consider the full-information setup when the entire reward vector is revealed to the learner at the end of each round. Apart from a technical result regarding the diameter of an auxiliary random process (Proposition 5 in the Appendix), the extension of the full-information policy to the bandit setting requires no substantially new ideas and will be dealt with in the following section. On

---

[1]This protocol includes conditionally deterministic policies, such as UCB, where $\boldsymbol{x}(t)$ is supported on only one arm.

[2]When we refer to the worst-case regret, we drop the argument $\boldsymbol{x}^*$ in parenthesis in the regret definition (3).

a high level, the BANDITQ policy first defines a *queueing* dynamics to take into account the gap between the target reward and the reward accrued by the policy for each arm so far. It then extends the *drift-plus-penalty* framework of Neely (2010, Chapter 4) to simultaneously achieve a small regret and meet the long-term constraints. However, to make this overall scheme work, we must adapt the asymptotic stochastic setting of Neely (2010) to the non-asymptotic adversarial setup with online information. This extension turns out to be highly non-trivial and requires new proof and algorithmic techniques, which are very different from that of the Max-Weight policy proposed by Neely (2010).

We associate a non-negative state variable $Q_i(t)$ to each protected arm $i \in \mathcal{P}$. Under the action of an online policy $\pi = \{\boldsymbol{x}(t)\}_{t \geq 1}$, the state variables evolve according to the following queueing dynamics, known as the Lindley recursion (Lindley, 1952):

$$Q_i(t) = \big(Q_i(t-1) + \lambda_i - r_i(t)x_i(t)\big)^+, \quad Q_i(0) = 0, \quad (6)$$

where we adopt the standard notation $(y)^+ \equiv \max(0, y)$. We set $Q_i(t) = 0, \forall t, \forall i \notin \mathcal{P}$. To get an intuition for Eq. (6), imagine that on every round $t$, a fixed deterministic amount of work $\lambda_i$ arrives at the queue $Q_i$. Then, under the action $\boldsymbol{x}(t)$ of an online policy, $\min(Q(t-1) + \lambda_i, r_i(t)x_i(t))$ amount of work departs from $Q_i$. It is intuitive that to stabilize the queues, the long-term service rates must be at least as large as the long-term arrival rates. Thus, any online policy stabilizing the queues would automatically satisfy the target rate requirements. However, since we are also interested in achieving a small regret, meeting the rate constraints alone is not enough (*c.f.* Huang et al. (2023)). Our online policy must also perform competitively in terms of the cumulative rewards against every feasible stationary action given by (1).

Towards this goal, let us first define the following quadratic potential function (*a.k.a.* Lyapunov function in the queueing theory parlance):

$$\Phi(t) = \sum_{i \in \mathcal{P}} Q_i^2(t). \quad (7)$$

We now have an upper bound on the change of potential under the action of a policy. From (6), we have

$$\begin{aligned} & Q_i^2(t) \\ \leq \ & \big(Q_i(t-1) + \lambda_i - r_i(t)x_i(t)\big)^2 \\ \leq \ & Q_i^2(t-1) + \lambda_i + x_i(t) + 2Q_i(t-1)(\lambda_i - r_i(t)x_i(t)), \end{aligned}$$

where, in the last inequality, we have used the fact that $0 \leq \lambda_i, r_i(t), x_i(t) \leq 1, \forall i, t$. Summing up the above inequality for each $i \in \mathcal{P}$, we have the following upper bound for the change of the potential on round $t$:

$$\Phi(t) - \Phi(t-1) \leq 2 + 2\sum_{i \in \mathcal{P}} Q_i(t-1)\big(\lambda_i - r_i(t)x_i(t)\big), \quad (8)$$

where we have used the fact that $\sum_i \lambda_i \leq 1, \sum_i x_i(t) \leq 1$, where the first inequality follows from the non-emptiness of $\Omega(\vec{\boldsymbol{\lambda}})$. Eqn. (8) suggests that running a MAB policy to maximize virtual cumulative rewards such that pulling the $i^{\text{th}}$ arm on round $t$ yields a virtual reward of $Q_i(t-1)r_i(t)$ will help minimize the change of the potential on round $t$ and, hence, meet the target rates. However, this does not explicitly take into account our other goal, namely to minimize the regret. To achieve both goals, motivated by the drift-plus-penalty framework of Neely (2010), we now define an instance of the standard online linear optimization (OLO) problem $\Xi$ with action set $\Delta_N$, where the surrogate reward of the $i^{\text{th}}$ arm on round $t$ is defined as:

$$r_i'(t) \equiv \big(Q_i(t-1) + V\big)r_i(t), \ \forall i \in [N]. \quad (9)$$

In the above, $V > 0$ is a hyper-parameter, to be fixed later, that depends only on the length of the horizon $T$. Intuitively, the surrogate reward vector $\boldsymbol{r}'(t)$ strikes a balance between attaining the target rates (through the first term) and achieving a small regret (through the second term). However, the definition of rewards (9) leads to two significant technical challenges in learning the surrogate rewards. First, due to the presence of the queue variables, the reward vectors $\boldsymbol{r}'(t)$ are not bounded *a priori*, which critically affects the regret bound for the surrogate problem $\Xi$. Second, although the original reward sequence $\{\boldsymbol{r}(t)\}_{t \geq 1}$ is i.i.d., the reward sequence $\{\boldsymbol{r}'(t)\}_{t \geq 1}$ for the problem $\Xi$ is *not* i.i.d. any more, again due to the presence of the queue variables, which are temporally correlated via Eqn. (6). The second difficulty prompts us to use an adversarial online learning policy for the auxiliary OLO problem $\Xi$.

**The BANDITQ policy:** The proposed BANDITQ policy can use any adaptive no-regret policy with a second-order regret bound for the auxiliary problem $\Xi$. This includes policies such as Online Gradient Ascent (OGA) with adaptive step sizes (Orabona, 2019) [3] and SQUINT (Koolen and Van Erven, 2015). To fix ideas, in this paper, we will use the OGA policy due to its simplicity. This online policy, which is closely related to the AdaGrad policy (Duchi et al., 2011), updates the sampling distribution on each round using the usual gradient step with an adaptive step size:

$$\boldsymbol{x}(t+1) \leftarrow \Pi_{\Delta_N}\left(\boldsymbol{x}(t) + \frac{\boldsymbol{r}'(t)}{\sqrt{2\sum_{\tau=1}^{t} \|\boldsymbol{r}'(\tau)\|_2^2}}\right). \quad (10)$$

In the above, the $\Pi_{\Delta_N}(\cdot)$ function, which denotes the Euclidean projection operator on the standard simplex $\Delta_N$, can be efficiently implemented in $O(N \log N)$ time (Wang and Carreira-Perpiñán, 2013). The complete BANDITQ policy in the full-information setting is summarized in Algorithm 1.

---

[3] Since ours is a maximization problem, we use a gradient ascent step rather than descent.

**Algorithm 1** BANDITQ Policy with full information

1: **Input:** Target reward rate vector $\vec{\lambda}$, Euclidean projection oracle $\Pi_{\Delta_N}(\cdot)$ onto the simplex $\Delta_N$.
2: $Q \leftarrow 0, x \leftarrow [1/N, 1/N, \ldots, 1/N], V \leftarrow \sqrt{T}, S \leftarrow 0.$ ▷ *Initialization*
3: **for each** round $t = 1 : T$ **do**
4:     Sample an arm $I_t$ from the distribution $x$.
5:     Observe the *entire* reward vector $r(t)$     ▷ *Full-information feedback*
6:     $Q_i = (Q_i + \lambda_i - r_i(t)x_i)^+, \forall i \in \mathcal{P}.$ ▷ *Updating the queue lengths*
7:     $r_i'(t) \leftarrow (Q_i + V)r_i(t), \forall i \in [N]$     ▷ *Computing the surrogate rewards*
8:     $S \leftarrow S + \|r'(t)\|^2.$ ▷ *Accumulating the norm of the past gradients*
9:     $x \leftarrow \Pi_{\Delta_N}\left(x + \frac{r'(t)}{\sqrt{2S}}\right)$     ▷ *Online gradient ascent*
10: **end for each**

In our analysis, we will use the following standard second-order regret bound achieved by the OGA policy with the above adaptive step sizes.

**Theorem 1** (Orabona (2019), Theorem 4.14). *Let $X \subseteq \mathbb{R}^d$ be a convex set with a finite Euclidean diameter $D$. Consider an arbitrary sequence of linear reward functions with gradients $\{g_t\}_{t\geq 1}$. Assume that the Online Gradient Ascent policy is run with step sizes[4] $\eta_t = \frac{D}{\sqrt{2\sum_{\tau=1}^t \|g_\tau\|_2^2}}, 1 \leq t \leq T$. Then the regret of the policy can be upper-bounded as follows:*

$$Regret_T \leq D\sqrt{2\sum_{t=1}^T \|g_t\|_2^2}. \quad (11)$$

It is important to note that the above bound is *scale-free*, *i.e.,* no *a priori* bounds on the gradients are needed for the above result (Putta and Agrawal, 2022; Hadiji and Stoltz, 2023). Specializing Theorem 1 to our surrogate problem $\Xi$, we obtain the following regret bound, which depends on the sequence of queue variables:

$$\begin{aligned}
\text{Regret}_t^{\Xi} &\leq 2\sqrt{\sum_{\tau=1}^t \sum_i (Q_i(\tau-1)+V)^2 r_i(t)^2} \\
&\leq 2\sqrt{2\sum_{\tau=1}^t \sum_i Q_i^2(\tau) + 2V\sqrt{2Nt}}. \quad (12)
\end{aligned}$$

In the above, we have used the fact that $0 \leq r_i(t) \leq 1, \forall t, i$, and the elementary inequalities $(a + b)^2 \leq 2(a^2 + b^2)$, $\sqrt{x+y} \leq \sqrt{x} + \sqrt{y}, x, y \geq 0$.

---

[4]Without any loss of generality, we set $\eta_t = 0$ if $g_t = 0$.

## 3.1 ANALYSIS

Unlike the analysis in Patil et al. (2021) and Cai et al. (2018), which proceed by constructing stochastic confidence intervals for the mean rewards of each arm, we directly make use of the regret bound (12) via an "adversarial-style" analysis, which critically makes use of a new *self-bounding* inequality derived below. Since the state variables $\{Q(t)\}_{t\geq 1}$ evolve according to the recursion (6), we do not immediately have an explicit control on the regret bound (12), which depends on the queue lengths. Hence, to bound the regret, we take an indirect approach. Fix any feasible distribution $x^* \in \Omega$. From Eq. (8), we have

$$\Phi(\tau) - \Phi(\tau-1) - 2V\sum_i r_i(\tau)x_i(\tau)$$
$$\leq 2 + 2\sum_i Q_i(\tau-1)\lambda_i -$$
$$2\sum_i \underbrace{(Q_i(\tau-1)+V)r_i(\tau)}_{r_i'(\tau)} x_i(\tau).$$

Summing up the above inequalities from $\tau = 1$ to $\tau = t$ and recalling that $\Phi(t) = \sum_i Q_i^2(t), \Phi(0) = 0$, we obtain

$$\sum_i Q_i^2(t) + 2\sum_{\tau=1}^t V\sum_i r_i(\tau)(x_i^* - x_i(\tau))$$
$$\leq 2t + 2\sum_{\tau=1}^t \sum_i Q_i(\tau-1)(\lambda_i - r_i(\tau)x_i^*) + 2\text{Regret}_t^{\Xi}, \quad (13)$$

where $\text{Regret}_t^{\Xi}$ denotes the worst-case regret for the surrogate problem (defined similarly as Eq. (3)). Note that, in the above, the regret bound on the RHS is random as it depends on the magnitude of the random process $\{Q(\tau)\}_\tau$. Let $\{\mathcal{F}_\tau\}_{\tau\geq 0}$ be the natural filtration generated by the sequence of rewards $\{r(\tau)\}_{\tau\geq 0}$. Taking expectations, we have the following set of inequalities for any benchmark distribution $x^* \in \Omega(\vec{\lambda})$:

$$\sum_i \mathbb{E}Q_i^2(t) + 2V\text{Regret}_t(x^*)$$
$$= \sum_i \mathbb{E}Q_i^2(t) + 2V\sum_{\tau=1}^t \mathbb{E}\sum_i r_i(\tau)(x_i^* - x_i(\tau))$$
$$\overset{(a)}{\leq} 2t + 2\sum_{\tau=1}^t \mathbb{E}\sum_i Q_i(\tau-1)(\lambda_i - x_i^*\mathbb{E}[r_i(\tau)|\mathcal{F}_{\tau-1}]) + 2\mathbb{E}[\text{Regret}_t^{\Xi}]$$
$$\overset{(b)}{\leq} 2t + 2\sum_{\tau=1}^t \mathbb{E}\sum_i Q_i(\tau-1)(\lambda_i - \mu_i x_i^*) + 2\mathbb{E}[\text{Regret}_t^{\Xi}]$$
$$\overset{(c)}{\leq} 2t + 2\mathbb{E}[\text{Regret}_t^{\Xi}]$$
$$\overset{(d)}{\leq} 2t + 4\sqrt{2\sum_{\tau=1}^t \sum_i \mathbb{E}Q_i^2(\tau)} + 4V\sqrt{2Nt}, \quad (14)$$

where in (a), we have taken the expectation of both sides of (13) with respect to the i.i.d. reward process $\{r(t)\}_{t\geq 1}$, and

used the law of iterated expectations; in (b), we have used the i.i.d. nature of the reward process; in (c) we have used the feasibility condition of the benchmark $\boldsymbol{x}^*$ from Eq. (1); in (d), we have used the second-order regret bound from Eq. (12) in conjunction with Jensen's inequality for the square root function. We emphasize that step $(d)$ is the *only* place where we use any property of the online learning subroutine. In other words, our reduction is *universal* in the sense that any online learning subroutine for $\Xi$, which could be very different from OGA but has a data-dependent regret bound similar to (11), can be used with BANDITQ .

Inequality (14) constitutes the key step in our analysis. It shows that the queue-length process $\{\boldsymbol{Q}(t)\}_{t \geq 1}$ possesses a *self-bounding* property in the sense that the expected queue-length squared at any round $t$ is bounded by the square root of the sum of expected queue-length squared up to round $t$ plus other auxiliary terms. The regret decomposition inequality (14) will be used to prove our main result in the full information setting.

**Theorem 2.** *The* BANDITQ *policy described in Algorithm 1 achieves the following regret and rate violation bounds:*

$$Regret_T = O(\max(\frac{T}{\sqrt{V}}, \sqrt{NT})), \mathbb{V}(T) = O(\sqrt{VT}).$$

*In particular, upon setting* $V = \sqrt{T}$, *we obtain*

$$Regret_T = O(\max(T^{3/4}, \sqrt{NT})), \ \mathbb{V}(T) = O(T^{3/4}).$$

The proof given below involves solving a non-linear sequential inequality to obtain a sublinear bound for the queue lengths. The resulting queue length bound is then used to control the regret.

*Proof.* First, we will derive a sublinear bound for the expected queue lengths under the BANDITQ policy. The rate violation and regret bounds will follow from this result.

**1 (a). Bounding the queue lengths:** Since the reward components are bounded in $[0, 1]$, using the fact that $\sum_i r_i(\tau)(x_i(\tau) - x_i^*) \leq 1, \forall \tau$, we have that $\text{Regret}_t(\boldsymbol{x}^*) \geq -t$. Hence, from Eq. (14), we have for all $t \geq 1$ :

$$\sum_i \mathbb{E}Q_i^2(t) \leq 2(V + 1)t + 4\sqrt{2 \sum_{\tau=1}^t \sum_i \mathbb{E}Q_i^2(\tau)} \\ + 4V\sqrt{2Nt}. \quad (15)$$

Hence, for any round $1 \leq \tau \leq t$, we have that

$$\sum_i \mathbb{E}Q_i^2(\tau) \leq 2(V + 1)t + 4\sqrt{2 \sum_{\tau=1}^t \sum_i \mathbb{E}Q_i^2(\tau)} + 4V\sqrt{2Nt}.$$

Summing up the above inequalities for all $\tau \in [1, t]$, we have

$$R^2(t) \leq 2(V + 1)t^2 + 4\sqrt{2N}Vt^{3/2} + 4\sqrt{2}tR(t).$$

where we have defined $R(t) \equiv \sqrt{\sum_{\tau=1}^t \sum_{i=1}^N \mathbb{E}Q_i^2(\tau)}$. Solving the above quadratic inequality in $R(t)$, we obtain

$$R(t) = O(t) + O(t\sqrt{V}) + O(N^{1/4}\sqrt{V}t^{3/4}) = O(t\sqrt{V}). \quad (16)$$

Plugging the above bound in (15), we have for each $i \in \mathcal{P}$ :

$$\mathbb{E}Q_i^2(t) = O(Vt) + O(t\sqrt{V}) + O(V\sqrt{N}t) = O(Vt)$$
$$\overset{\text{(Jensen's ineq.)}}{\Longrightarrow} \mathbb{E}Q_i(t) = O(\sqrt{Vt}). \quad (17)$$

**1 (b). Bounding the rate violation penalty** $\mathbb{V}(T)$**:** Upon expanding (6), we obtain the following well-known representation for the Lindley recursion (Asmussen, 2003, pp. 92):

$$Q_i(t) = \sup_{1 \leq \tau \leq t} (0, \lambda_i\tau - \sum_{z=t-\tau+1}^t r_i(z)x_i(z)), \ \forall i \in \mathcal{P}. \quad (18)$$

Combining Eq. (18) with the bound (17), we can bound the constraint violation penalty as

$$\mathbb{V}(T) \leq \max_{i \in \mathcal{P}} \mathbb{E}Q_i(T) = O(\sqrt{VT}).$$

**2. Bounding the regret:** Substituting (16) into the inequality (14) and using the fact that $Q_i^2(T) \geq 0, \forall i, t$, we have for any $\boldsymbol{x}^* \in \Omega$ :

$$2V\text{Regret}_T(\boldsymbol{x}^*) \leq O(T) + O(T\sqrt{V}) + O(V\sqrt{NT}).$$

This yields the following regret bound

$$\begin{aligned} \text{Regret}_T(\boldsymbol{x}^*) &= O(\frac{T}{V}) + O(\frac{T}{\sqrt{V}}) + O(\sqrt{NT}) \\ &= O(\max(\frac{T}{\sqrt{V}}, \sqrt{NT})). \end{aligned}$$

$\square$

**Remarks:** It may appear from the statement of Theorem 2 that BANDITQ achieves a sub-optimal $O(T^{3/4})$ regret bound even for the standard regret minimization problem with no specified target reward rates, *i.e.,* $\boldsymbol{\lambda} = \boldsymbol{0}$. However, as we show in Section B of the Appendix, the BANDITQ policy actually achieves the optimal instance-independent $O(\sqrt{T})$ regret bound for both full-information and bandit feedback settings for $\boldsymbol{\lambda} = \boldsymbol{0}$.

As an immediate corollary of Theorem 2, the following result shows that under the action of the BANDITQ policy, the target reward accrual rates are met asymptotically for each arm $i \in \mathcal{P}$ :

**Proposition 1.** *Upon setting* $V = \sqrt{T}$, *for any interval* $\mathcal{I} \subseteq [T]$ *such that* $T^{3/4} = o(|\mathcal{I}|)$, *the* BANDITQ *policy in the full-information setting yields:*

$$\liminf_{|\mathcal{I}| \to \infty} |\mathcal{I}|^{-1}\mathbb{E}\sum_{t \in \mathcal{I}} r_i(t)x_i(t) \geq \lambda_i, \ \forall i \in \mathcal{P}.$$

See Appendix C for the proof. Although Theorem 2 gives an $O(T^{3/4})$ regret bound compared to the minimax regret bound of $O(\sqrt{T \log N})$ for the unconstrained problem (Zhao and Chen, 2019, Theorem 4), the next result shows that the proposed BANDITQ policy achieves a substantially stronger $O(\sqrt{T})$ bound for the *average* regret, where the regret is averaged over the entire time horizon $T$.

**Proposition 2.** *In the full information setting, under the BANDITQ policy with $V = \sqrt{T}$, we have $\frac{1}{T}\sum_{t=1}^{T} Regret_t(\boldsymbol{x}^*) = O(\sqrt{NT})$, for any $\boldsymbol{x}^* \in \Omega$ and $\mathbb{V}(T) = O(T^{3/4})$.*

*Proof.* Define $S_t^2 \equiv \sum_i \mathbb{E}Q_i^2(t)$. From Eq. (13), for all $t \in [T]$, we have

$$
\begin{aligned}
S_t^2 + 2V\text{Regret}_t(\boldsymbol{x}^*) &\leq 2t + 4\sqrt{2\sum_{\tau=1}^{t} S_\tau^2} + 4V\sqrt{2Nt} \\
&\leq 2T + 4\sqrt{2\sum_{\tau=1}^{T} S_\tau^2} + 4V\sqrt{2NT}.
\end{aligned}
$$

Summing up the above inequalities from $t = 1$ to $t = T$ and defining $z_T \equiv \sqrt{\sum_{\tau=1}^{T} S_\tau^2}$, we obtain

$$
z_T^2 - 4Tz_T + 2V\sum_{t=1}^{T} \text{Regret}_t(\boldsymbol{x}^*) \leq 2T^2 + 4V\sqrt{2N}T^{3/2}. \quad (19)
$$

Upon completing the square, we have $z_T^2 - 4Tz_T = (z_T - 2T)^2 - 4T^2 \geq -4T^2$. Hence, from (19), we conclude that:

$$
\frac{1}{T}\sum_{t=1}^{T} \text{Regret}_t(\boldsymbol{x}^*) \leq 3\frac{T}{V} + 2\sqrt{2NT}.
$$

The final result follows upon setting $V = \sqrt{T}$. $\square$

Finally, if one is only interested in achieving the target rate vector $\vec{\lambda}$ while completely disregarding the regret, the following Proposition shows that the queue-length bound, and hence, the rate violation penalty given in Proposition 2 can be further improved to $O(\sqrt{T})$ upon setting $V = 0$.

**Proposition 3.** *The cumulative constraint violation under the BANDITQ policy in the full-information setting with $V = 0$ can be bounded as follows:*

$$
\mathbb{V}(T) \leq \max_i \mathbb{E}Q_i(T) \leq 6\sqrt{T}.
$$

*Proof.* From Eq. (14), we have for any fixed $t$ and any $1 \leq \tau \leq t$:

$$
\sum_i \mathbb{E}Q_i^2(\tau) \leq 2t + 4\sqrt{2\sum_{\tau=1}^{t}\sum_i \mathbb{E}Q_i^2(\tau)} \quad \forall t \geq 1, \forall i. \quad (20)
$$

Summing up the above inequalities for $1 \leq \tau \leq t$ and defining $z_t^2 \equiv \sum_{\tau=1}^{t}\sum_i \mathbb{E}Q_i^2(\tau)$, we have

$$
z_t^2 \leq 2t^2 + 4\sqrt{2}tz_t.
$$

Solving the above quadratic inequality, we conclude that

$$
\sqrt{\sum_{\tau=1}^{t}\sum_i \mathbb{E}Q_i^2(\tau)} = z_t \leq 6t.
$$

Substituting the above bound in (20) and using Jensen's inequality, we conclude that $\mathbb{E}Q_i(t) \leq 6\sqrt{t}, \forall i \in [N]$. $\square$

**Sharper regret bound under a monotonicity assumption:** The regret and constraint violation bounds derived above hold unconditionally. We now show that the BANDITQ policy achieves the minimax optimal $O(\sqrt{T})$ regret under a mild monotonicity assumption on the queue length sequence stated below.

**Assumption 1** (Monotonicity in expectation)**.** *Under the action of the chosen OLO subroutine, the sequence of variables $Q^2(t) \equiv \sum_i \mathbb{E}Q_i^2(t), t \geq 1$ are non-decreasing in $t$.*

**Theorem 3.** *Under Assumption 1, the regret of the BANDITQ policy in the full-information setting is bounded as*

$$
Regret_t \leq \frac{5t}{V} + 2\sqrt{2Nt}, \ 1 \leq t \leq T.
$$

*In particular, with $V = \sqrt{T}$, we have $Regret_t = O(\sqrt{Nt})$ for any $t \in [T]$.*

See Appendix D for the proof. Assumption 1 is related to a stochastic monotonicity assumption. Many closely related Markov chains, *e.g.,* the birth-death chain, which is a continuous-time model of a queue with zero initial states, are known to be stochastically monotone (Ross, 1995, Proposition 9.2.4) (Van Doorn, 1980, Theorem 6.1), (Keilson and Kester, 1977).

## 4 BANDITQ POLICY WITH BANDIT FEEDBACK

Under bandit feedback, only the reward of the selected arm, *i.e.,* $r_{I_t}(t)$, is revealed to the policy at the end of round $t$. The reader should compare this with the full-information setup where the entire reward vector $\boldsymbol{r}(t)$ is revealed irrespective of the action. To deal with the resulting in the *exploration-vs-exploitation* trade-off in the limited information setup, we replace the full-information OGA policy (10) with an adversarial MAB policy, proposed recently by Putta and Agrawal (2022), that enjoys a *scale-free* second-order regret bound similar to Eq. (11). Their *Follow-the-regularized-leader* (FTRL)-based MAB policy uses the standard inverse propensity score to estimate the reward vectors and employs a log-barrier regularizer in the FTRL algorithm with a carefully chosen learning rate schedule. The arms are finally selected by mixing a uniform exploration component with the distribution obtained from the FTRL algorithm. For

completeness, we describe the BANDITQ policy in the bandit information setting in Appendix G. Putta and Agrawal (2022) showed that their proposed MAB policy works for *any* real loss vector (unlike, *e.g.,* EXP3, which requires non-negative losses) and enjoys the following scale-free adaptive regret bound.

**Theorem 4** (Putta and Agrawal (2022)). *MAB Algorithm 1 of Putta and Agrawal (2022), when run with the oblivious linear reward sequence with coefficient vectors $\{\boldsymbol{g}_t\}_{t=1}^T$, enjoys the following scale-free regret bound:*

$$Regret_T = \tilde{O}\left(\sqrt{N \sum_{t=1}^T \|\boldsymbol{g}_t\|_2^2} + \max_{t \in [T]} \|\boldsymbol{g}_t\|_\infty \sqrt{NT}\right). \quad (21)$$

It can be seen that the only essential difference between the above expression and that of the OGA regret bound in Eq. (11) is the presence of the additional term $\tilde{O}(\max_{t \in [T]} \|\boldsymbol{g}_t\|_\infty \sqrt{NT})$ in the former. With a more careful analysis using martingales, our previous arguments go through with minimal changes. We now outline the main differences between the full information and the bandit setup.

**Notation:** Let us encode the index of the selected arm $I_t$ on round $t$ by the one-hot encoded vector $\boldsymbol{X}(t) = (X_1(t), X_2(t), \dots, X_N(t)) \in \{0,1\}^N$, where $X_i(t) = \mathbb{1}(I_t = i), \forall i$. Thus, if $x_i(t)$ denotes the conditional probability that the $i^{\text{th}}$ arm is pulled, we have $\mathbb{P}(X_i(t) = 1|\mathcal{F}_{t-1}) = 1 - \mathbb{P}(X_i(t) = 0|\mathcal{F}_{t-1}) = x_i(t)$ and $\mathbb{E}(X_i(t)|\mathcal{F}_{t-1}) = x_i(t), \forall i, t$.

**Queueing recursion and the auxiliary MAB problem:** Note that the queueing recursion (6) for the full-feedback setting does not work in the case of Bandit feedback because the rewards of the unobserved arms are not revealed. However, it is straightforward to modify the recursion (6) by replacing the sampling probabilities $\boldsymbol{x}(t)$ with the corresponding random realizations $\boldsymbol{X}(t)$. Hence, in the bandit setting, the queueing evolution for the $i^{\text{th}}$ arm reads:

$$Q_i(t) = \left(Q_i(t-1) + \lambda_i - r_i(t)X_i(t)\right)^+, \ Q_i(0) = 0. \quad (22)$$

Eq. (22) is well-defined in the bandit feedback setting as $X_i(t) = 0$ if $i \neq I_t$. Hence, the recursion (22) does not depend on the reward of any arm which was not played. Next, analogous to the full-information setting (Eq. (9)), the BANDITQ policy defines an instance of an adversarial MAB problem $\Xi^{\text{BANDIT}}$ where the surrogate reward of the $i^{\text{th}}$ arm on round $t$ is defined as:

$$r_i'(t) \equiv \left(Q_i(t-1) + V\right)r_i(t), \ \forall i \in [N]. \quad (23)$$

As before, the surrogate rewards are not bounded *a priori* due to the presence of the queueing variables.

## 4.1 ANALYSIS

As before, the components of the surrogate reward gradients are given by $\boldsymbol{g}_{t,i} = r_i'(t) = \left(Q_i(t-1) + V\right)r_i(t)$. Using

the quadratic potential function $\Phi(\cdot)$ defined in Eq. (7) and working identically up to step (c) of Eq. (14), we derive the following self-bounding inequality:

$$\sum_i \mathbb{E}Q_i^2(t) + 2V \text{Regret}_t(\boldsymbol{x}^*)$$

$$\leq 2t + 2\mathbb{E}\left[\text{Regret}_t^{\Xi^{\text{Bandit}}}\right]$$

$$\overset{(a)}{\leq} 2t + \tilde{O}\Bigg(\sqrt{N \sum_{\tau=1}^t \sum_i \mathbb{E}Q_i^2(\tau)} + NV\sqrt{t} +$$

$$V\sqrt{Nt} + \sqrt{Nt}\mathbb{E}\left[\max_{i,\tau \in [t]}(Q_i(\tau))\right]\Bigg) \quad (24)$$

$$\overset{(b)}{\leq} 2t + \tilde{O}\Bigg(\sqrt{N \sum_{\tau=1}^t \sum_i \mathbb{E}Q_i^2(\tau)} + NV\sqrt{t} + \sqrt{N}t^{3/2}\Bigg),$$

$$(25)$$

where, in step (a), we have used the regret bound from Theorem 4, and in step (b), we have used the trivial bound $Q_i(t) \leq t, \forall t \in [T], \forall i$. The following theorem gives the performance of the BANDITQ policy with bandit feedback.

**Theorem 5.** *In the bandit feedback setting, the BANDITQ policy achieves the following regret and target rate violation bounds:*

$$Regret_T = \tilde{O}(\max(\frac{T\sqrt{N}}{\sqrt{V}}, \frac{N^{3/4}T^{5/4}}{V}, N\sqrt{T})).,$$

$$\mathbb{V}(T) = \tilde{O}(\max(\sqrt{VT}, N^{1/4}T^{3/4})).$$

*In particular, upon setting $V = \sqrt{T}$, we obtain*

$$Regret_T = O(N^{3/4}T^{3/4}), \ \mathbb{V}(T) = \tilde{O}(N^{1/4}T^{3/4}).$$

Compared to the full-information setting, the proof in the bandit setting uses a more sophisticated Martingale-based argument to control the maximum of the queueing process for bounding the second term in the regret expression (21). To simplify the exposition, the proof of Theorem 5 is broken into three interrelated propositions. We begin our analysis by first deriving a sublinear bound for $\mathbb{E}Q_i^2(t)$.

**Proposition 4.** *Under the action of the BANDITQ policy with bandit feedback, we have*

$$\mathbb{E}Q_i^2(t) = \tilde{O}(\max(Vt, \sqrt{N}t^{3/2})), \forall i, t.$$

*Hence, using Jensen's inequality, we have $\mathbb{V}(T) = \tilde{O}(\max(\sqrt{VT}, N^{1/4}T^{3/4}))$.*

*Proof.* Recall that from Eqn. (25) we have:

$$\sum_i \mathbb{E}Q_i^2(t) + 2V \text{Regret}_t(\boldsymbol{x}^*) \leq 2t +$$

$$\tilde{O}\Bigg(\sqrt{N \sum_{\tau=1}^t \sum_i \mathbb{E}Q_i^2(\tau)} + NV\sqrt{t} + \sqrt{N}t^{3/2}\Bigg).$$

Using the fact that $r_i(t) \leq 1, \forall i, t$, we have $\text{Regret}_t(x^*) \geq -t$. Hence, from the above, we obtain

$$\sum_i \mathbb{E}Q_i^2(t) \leq 2(V+1)t +$$

$$\tilde{O}\Big(\sqrt{N\sum_{\tau=1}^{t}\sum_i \mathbb{E}Q_i^2(\tau)} + NV\sqrt{t} + \sqrt{N}t^{3/2}\Big), \quad (26)$$

which resembles Eqn. (15) in the full-information setting. Defining $R(t) \equiv \sqrt{\sum_{\tau=1}^{t}\sum_{i=1}^{N}\mathbb{E}Q_i^2(\tau)}$ and working similarly as in the full-information setting, we have the following quadratic inequality:

$$\begin{aligned} R^2(t) &\leq 2(V+1)t^2 + \\ &\quad \tilde{O}\big(\sqrt{N}tR(t) + NVt^{3/2} + \sqrt{N}t^{5/2}\big) \\ \implies R(t) &= \tilde{O}\big(\max(t\sqrt{V}, N^{1/4}t^{5/4})\big). \quad (27) \end{aligned}$$

Substituting the above bound in (26), we conclude that for each $i \in [N]$:

$$\mathbb{E}Q_i^2(t) = \tilde{O}(\max(Vt, \sqrt{N}t^{3/2})).$$

$\square$

The next proposition establishes a sublinear bound to the diameter $\mathbb{E}\big[\max_{i,t\in[T]}Q_i(t)\big]$, which appears on the RHS of (24).

**Proposition 5.** *Under the action of the* BANDITQ *policy, for any round $T \geq 1$, we have the following bound for the expected maximum of the queueing processes*

$$\mathbb{E}\big[\max_{i,t\in[T]}Q_i(t)\big] = \tilde{O}(\max(\sqrt{VT}, N^{1/4}T^{3/4})).$$

The proof of Proposition 5 is technical and is given in Section E in the Appendix. Combining the above two results, the following proposition gives the worst-case regret bound for the BANDITQ policy under the bandit feedback.

**Proposition 6.** *The worst-case regret of the* BANDITQ *policy under the bandit feedback is bounded as*

$$\text{Regret}_T = \tilde{O}(\max(\frac{T\sqrt{N}}{\sqrt{V}}, \frac{N^{3/4}T^{5/4}}{V}, N\sqrt{T})).$$

*Proof.* From Eqn. (24), we have

$$\sum_i \mathbb{E}Q_i^2(T) + 2V\text{Regret}_T(x^*) \leq 2T + \tilde{O}\Big(\sqrt{N}R(T) +$$

$$NV\sqrt{T} + V\sqrt{NT} + \sqrt{NT}\mathbb{E}\big[\max_{i,\tau\in[T]}(Q_i(\tau))\big]\Big), \quad (28)$$

where $R(T) \equiv \sqrt{\sum_{\tau=1}^{T}\sum_{i=1}^{N}\mathbb{E}Q_i^2(\tau)}$. Plugging in the upper bound for $R(T)$ from Eqn. (27) and the diameter of the queueing process from Proposition 5, we obtain:

$$2V\text{Regret}_T(x^*) = \tilde{O}(\max(T\sqrt{NV}, NV\sqrt{T}, N^{3/4}T^{5/4})).$$

Hence,

$$\text{Regret}_T(x^*) = \tilde{O}(\max(\frac{T\sqrt{N}}{\sqrt{V}}, \frac{N^{3/4}T^{5/4}}{V}, N\sqrt{T})).$$

$\square$

Proposition 4 and Proposition 6, taken together, establish Theorem 5.

Following exactly the same arguments, the result in Proposition 1 can be shown to hold in the bandit feedback setting as well. Finally, as in the full-information setting, we now discuss the case when one is only interested in satisfying the target rate constraints while disregarding the accrued rewards. The following proposition gives a bound on the cumulative violation in the bandit setting.

**Proposition 7.** *Setting $V = 0$, the cumulative constraint violation under the* BANDITQ *policy in the bandit setting can be bounded for any $T \geq 1$ as follows:*

$$\mathbb{V}(T) \leq \max_i \mathbb{E}Q_i(T) = \tilde{O}(N^{3/8}T^{5/8}).$$

The above bound is slightly worse compared to the $O(\sqrt{T})$ bound in the full-information setting (Proposition 3). See Section F in the Appendix for the proof of Proposition 7.

**Remarks:** Technically, the scale-free regret bound given in Theorem 4 was derived for *oblivious* adversaries, which fixes the entire sequence of reward vectors at $t = 0$. However, in our case, the surrogate reward vector $r'(t)$ in Eqn. (23) is determined by the past actions of the policy through the variable $Q(t)$. To see why we can still use the regret bound (21), note that the surrogate reward $r'(t)$ does not depend on the current action $X(t)$. Hence, we can invoke the regret bound for an imaginary adversary that decides the reward vector $r'(t)$ at the end of round $t-1$. Since the reward on round $t$ does not affect the previous actions of the policy, the regret bound (21) applies to our problem.

## 5 EXPERIMENTS

**Simulation Setup:** We consider a problem instance with $N = 5$ arms and $k = 2$ protected classes consisting of the first and the second arm. We arbitrarily set the mean reward vector of the arms to $\mu = (0.335, 0.203, 0.241, 0.781, 0.617)$, and the target reward rates for the first and the second arm to $\lambda_1 = 0.167$ and $\lambda_2 = 0.067$ respectively. From Eq. (2), it can be verified that the required rates are feasible for this problem. Clearly, Arm # 4 is the most rewarding among the five arms. We simulate the BANDITQ policy for $T = 2 \times 10^6$ rounds upon setting the parameter $V = \sqrt{T} \approx 1414$. We write a custom optimizer, described in Appendix H, to efficiently implement the optimization subroutines. The simulation code has been made publicly available (Sinha, 2024).

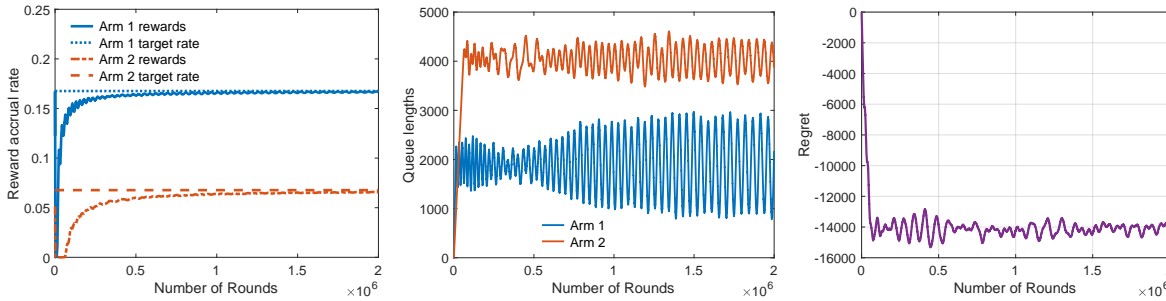

Figure 1: Reward accrual rates in the full-information setting

Figure 2: Queue lengths in the full-information setting

Figure 3: Regret of BANDITQ in the full-information setting

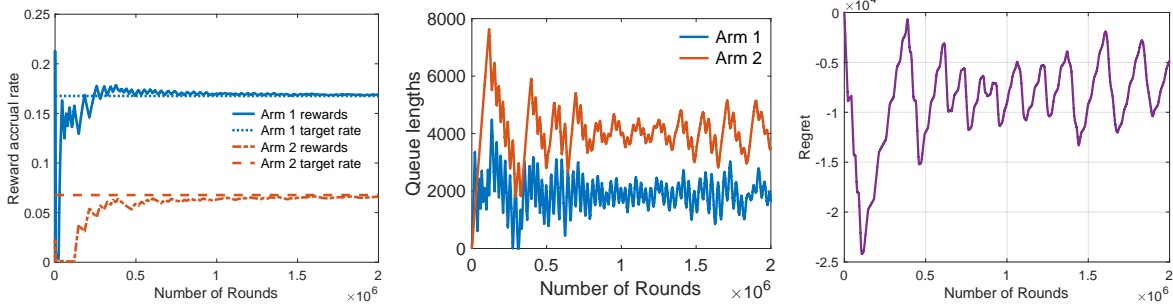

Figure 4: Reward accrual rates in the bandit feedback

Figure 5: Queue lengths in the bandit feedback setting

Figure 6: Regret of BANDITQ in the bandit feedback setting

**Discussion:** Figures 8, 9, and 10 show the performance of the BANDITQ policy in the full-information set-up. Figure 8 shows that the protected arms, Arm 1 and Arm 2, asymptotically meet their target rates. Observe that since both Arm 1 and Arm 2 have sub-optimal expected rewards, they would have received asymptotically zero reward rates under the action of an unfair prediction policy, such as UCB. Figure 9 shows the evolution of the queue length variables, and Figure 10 shows the regret of the BANDITQ policy in the full-information setting. Negative values of the regret suggest that the cumulative reward of the BANDITQ policy exceeds the reward achieved by the static benchmark policy, which is forced to take actions from the restricted set $\Omega$ on *all* rounds - a constraint that the BANDITQ policy does not need to respect on every round. Figures 11, 12, and 13 show the corresponding plots in the bandit feedback setting. As expected, in the case of bandit feedback, the variables exhibit greater empirical variance compared to their full-information counterpart due to the limited availability of information. However, the BANDITQ policy achieves the target rates in this case as well. See Section I.2 in the Appendix for a similar experiment with $N = 1000$ arms.

**Additional experiments:** A comparison of the BANDITQ policy with a UCB-based oracle policy, proposed by Li et al. (2019), has been given in Appendix I.1. The oracle is assumed to know a feasible fraction of pulls to achieve the target rates. The plot in Figure 7 shows that the proposed BANDITQ policy achieves more cumulative rewards com-

pared to the oracle policy as it decides its actions adaptively.

## 6   CONCLUSION AND OPEN PROBLEMS

In this paper, we proposed a black-box reduction from the fair bandits problem to the unconstrained bandits problem and bounded the regret and cumulative target rate violations. Since we use adversarial MAB policies as subroutines, it is reasonable to conjecture that the proposed BANDITQ policy would work in the adversarial setting as well. Substantiating this statement would be an interesting research direction (Sinha and Vaze, 2023). Improving the regret and rate violation bounds by, *e.g.,* working with a different Lyapunov function would be practically useful (Sinha and Vaze, 2024). Finally, coming up with sharper instance-dependent regret bounds would be interesting as well.

## 7   ACKNOWLEDGEMENT

The work was supported in part by a Google India faculty Research award and in part by a US-India NSF-DST collaborative grant coordinated by IDEAS-Technology Innovation Hub (TIH) at the Indian Statistical Institute, Kolkata. The author gratefully acknowledges a lively discussion with Prof. Anurag Kumar from IISc while formulating the problem.

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

# Appendix for
# BANDITQ: Fair Bandits with Guaranteed Rewards

**Abhishek Sinha**[1]

[1] School of Technology and Computer Science, Tata Institute of Fundamental Research, Mumbai 400005, India
abhishek.sinha@tifr.res.in

## A  ON THE FEASIBILITY ASSUMPTION

Throughout the paper, we assume that the target rate vector $\vec{\lambda}$ is feasible. In practice, we can ensure the feasibility by estimating the expected rewards from past data and requiring that condition (2) is strictly satisfied with a reasonable margin. To put it quantitatively, let $\hat{\boldsymbol{\mu}}$ be the estimated expected reward vector where it is known that $\|\hat{\boldsymbol{\mu}} - \boldsymbol{\mu}\|_\infty \le \epsilon$, for a small error bound $\epsilon \ge 0$. Then, for the required reward rate vector $\vec{\lambda}$ to be feasible, using the first-order Taylor's series expansion, it is sufficient that:

$$\sum_i \frac{\lambda_i}{\hat{\mu}_i} + \epsilon \sum_i \frac{\lambda_i}{\hat{\mu}_i{}^2} \le 1. \tag{29}$$

Although the estimated mean rewards can reasonably be used for determining the feasibility of the required reward rates, they cannot possibly be used for the online selection of the arms with no regret, as even a small constant error in the estimated rewards may lead to a linear regret.

## B  $O(\sqrt{T})$ REGRET OF THE BANDITQ POLICY WITH NO TARGET RATES

We now consider the classical and special case when there are no specific target rates for any of the arms, *i.e.,* $\lambda_i = 0, \forall i$. Hence, from Eqn. (6), we have that $Q_i(t) = 0, \forall i, t$. Furthermore, with $\boldsymbol{\lambda} = \mathbf{0}$, the comparator class $\Omega$ coincides with the set of all probability distributions over $N$ arms ($\Delta_N$). We have the following result

**Proposition 8.** *With no pre-specified target reward rates,* i.e., $\boldsymbol{\lambda} = \mathbf{0}$, *the BANDITQ policy achieves regret bounds of* $O(\sqrt{Nt})$ *and* $\tilde{O}(N\sqrt{t})$ *for the full-information and bandit feedback settings, respectively.*

Intuitively, the above result can be understood from the fact that, in this case, the surrogate rewards $\boldsymbol{r}'(t)$ of the BANDITQ policy is simply a scaled version of the original rewards $\boldsymbol{r}(t)$. See below for a formal proof.

**Proof:**

**Full-information setting:**  From the regret decomposition inequality (14), we have that

$$2V \text{Regret}_t(x^*) \le 2t + 4V\sqrt{2Nt}.$$

Setting $V = \sqrt{T}$, we have that

$$\text{Regret}_t(x^*) \le \frac{t}{V} + 2\sqrt{2Nt} = O(\sqrt{Nt}).$$

**Bandit information setting:** The proof is almost identical to the full-information case. Setting $Q_i(t) = 0, \forall i, t$, in the regret decomposition inequality (24), we have that

$$2V \operatorname{Regret}_t(x^*) \leq 2t + \tilde{O}(NV\sqrt{t} + V\sqrt{Nt}).$$

Setting $V = \sqrt{T}$, the above yields

$$\operatorname{Regret}_t(x^*) = \tilde{O}(N\sqrt{t}).$$

## C PROOF OF PROPOSITION 1

Using Proposition 2, we have that $\mathbb{E}Q_i(t) \overset{\text{(Jensen's ineq.)}}{\leq} \sqrt{\mathbb{E}Q_i^2(t)} = O(N^{1/4}T^{3/4}), \ \forall i \in \mathcal{P}, t \in [T]$. Let $\mathcal{I} \subseteq [1, T]$ be any sub-interval of length $l = |\mathcal{I}|$. Substituting the above bound in Eq. (18), we have for any $i \in \mathcal{P}$:

$$\inf_{\mathcal{I}} \mathbb{E}\big( 1/|\mathcal{I}| \sum_{z \in \mathcal{I}} r_i(z)x_i(z) \big) \geq \lambda_i - O(\frac{N^{1/4}T^{3/4}}{l}),$$

which gives a finite-time guarantee for the expected reward accrual rate for each arm in the protected set $\mathcal{P}$. Hence, as long as $T^{3/4}/l \to 0$, we have

$$\liminf_{|\mathcal{I}| \to \infty} |\mathcal{I}|^{-1} \mathbb{E}\big[ \sum_{t \in \mathcal{I}} r_i(t)x_i(t) \big] \geq \lambda_i, \ \forall i \in \mathcal{P}.$$

## D PROOF OF THEOREM 3

Let $x^*$ be an optimal fixed feasible randomized action. From Eqn. (14), we have that

$$2V \operatorname{Regret}_t(x^*) \leq 4\sqrt{2 \sum_{\tau=1}^{t} \sum_i \mathbb{E}Q_i^2(\tau) - \sum_i \mathbb{E}Q_i^2(t)} + 2t + 4V\sqrt{2Nt}.$$

Define $Q^2(\tau) = \sum_i \mathbb{E}Q_i^2(\tau), \forall \tau \geq 1$. Using the monotonicity assumption 1, the above inequality yields

$$
\begin{aligned}
2V \operatorname{Regret}_t(x^*) \quad &\leq \quad \underbrace{4\sqrt{2t}Q(t) - Q^2(t)}_{(A)} + 2t + 4V\sqrt{2Nt} \\
&\overset{(a)}{\leq} \quad 10t + 4V\sqrt{2Nt}.
\end{aligned}
$$

where in (a), we have upper-bounded the quadratic (A), which is of the form $ax - x^2$, by $a^2/4 \equiv 8t$. Hence, we have

$$\operatorname{Regret}_t \leq \frac{5t}{V} + 2\sqrt{2Nt}.$$

## E PROOF OF PROPOSITION 5

*Proof.* Using Eq. (13) and the fact that $|r_i(t)| \leq 1, \forall i, t$, we have the following sample-path wise bound on the square of the queue lengths:

$$
\begin{aligned}
\sum_i Q_i^2(t) \quad &\leq \quad 2(V+1)t + 2 \sum_{\tau=1}^{t} \sum_i Q_i(\tau-1)\big( \lambda_i - r_i(\tau)x_i^* \big) + 2\operatorname{Regret}_t^\Xi \\
&\leq \quad 2(V+1)t + \tilde{O}\big( \sqrt{N \sum_{\tau=1}^{t} \sum_i Q_i^2(\tau)} + NV\sqrt{t} + \sqrt{Nt} \max_{i,\tau \in [1,t]} Q_i(\tau) \big) + 2 \sum_i M_t^i, \quad (30) \\
&\overset{(a)}{\leq} \quad 2(V+1)t + \tilde{O}\big( \sqrt{N \sum_{\tau=1}^{t} \sum_i Q_i^2(\tau)} + NV\sqrt{t} + \sqrt{N}t^{3/2} \big) + 2 \sum_i M_t^i, \quad (31)
\end{aligned}
$$

where in the above, we have substituted the upper bound to the regret of the surrogate problem from Eq. (21) (as in Eqn. (25)), used the fact that $Q_{(\tau)} \le \tau$, and for each $i \in [N]$, we have defined the stochastic process $\{M_t^i\}_{t \ge 1}$ as follows:

$$M_t^i = \sum_{\tau=1}^{t} Q_i(\tau - 1)\big(\lambda_i' - r_i(\tau)x_i^*\big), \ t \ge 1. \tag{32}$$

where $\lambda_i' \overset{(\text{def.})}{=} x_i^* \mathbb{E} r_i(\tau) = x_i^* \mu_i \ge \lambda_i$. Taking the maximum of both sides with respect to all rounds $t \in [T]$ for some $T \ge 1$, we have

$$\max_{i,t \in [T]} Q_i^2(t) \le 2VT + \tilde{O}\Big(\sqrt{N \sum_{\tau=1}^{T} \sum_i Q_i^2(\tau)} + NV\sqrt{T} + \sqrt{N}T^{3/2}\Big) + 2 \sum_i \max_{t \in [T]} M_t^i.$$

Taking the expectation of both sides of the above inequality, we obtain

$$
\begin{aligned}
\mathbb{E}\Big[\max_{i,t \in [T]} Q_i^2(t)\Big] &\le 2VT + \tilde{O}\Big(\sqrt{N \sum_{\tau=1}^{T} \sum_i \mathbb{E}Q_i^2(\tau)} + \sqrt{N}T^{3/2}\Big) + 2 \sum_i \mathbb{E}\Big[\max_{t \in [T]} M_t^i\Big] \\
&\overset{(a)}{\le} 2VT + \tilde{O}\big(\max(T\sqrt{V}, N^{1/4}T^{5/4})\big) + \tilde{O}(\sqrt{N}T^{3/2}) + 2 \sum_i \mathbb{E}\Big[\max_t M_t^i\Big] \quad (33) \\
&= \tilde{O}(\max(VT, \sqrt{N}T^{3/2})) + 2 \sum_i \mathbb{E}\Big[\max_t M_t^i\Big], \quad (34)
\end{aligned}
$$

where in step (a), we have used the bound for $R(T)$ from Eqn. (27). Next, we claim that each of the processes $\{M_t^i\}_{t \ge 1}$ is a zero-mean martingale process with respect to the natural filtration $\{\mathcal{F}_\tau\}_{\tau \ge 1}$. This follows from the definition (32) as $Q_i(\tau - 1) \in \mathcal{F}_{\tau-1}$ is pre-visible, and the random variable $r_i(\tau)$ is independent of $\mathcal{F}_{\tau-1}$ s.t. $\mathbb{E}(\lambda_i' - r_i(\tau)x_i^*) = 0$. Using the $L^2$ maximum inequality for Martingales (Durrett, 2019, Theorem 4.4.4), (Doob, 1953, Theorem 3.4), (Dubins and Schwarz, 1988), we have

$$\mathbb{E}[\max_{t \in [T]} M_t^i] \le 2\sqrt{\mathbb{E}(M_T^i)^2}. \tag{35}$$

Since $\{M_t\}_{t \ge 1}$ is a zero-mean martingale sequence, using the Pythagorean formula for martingales (Williams, 1991, Eq. (b), Section 12.1) and the fact that $|\lambda_i' - r_i(\tau)x_i^*| \le 1$, we have

$$
\begin{aligned}
\mathbb{E}(M_T^i)^2 &\le \sum_{\tau=1}^{T} \mathbb{E}Q_i^2(\tau - 1) \quad (36) \\
&\le R^2(T),
\end{aligned}
$$

where we have defined $R(T) \equiv \sqrt{\sum_{\tau=1}^{T} \sum_{i=1}^{N} \mathbb{E}Q_i^2(\tau)}$. Combining the above with Eq. (33), we obtain the desired bound for the diameter of the queueing process:

$$\mathbb{E}\Big[\max_{i,t \in [T]} Q_i^2(t)\Big] \le \tilde{O}(\max(VT, \sqrt{N}T^{3/2})) + O(R(T)) = \tilde{O}(\max(VT, \sqrt{N}T^{3/2})),$$

where we have again used the bound for $R(T)$ from Eqn. (27). The result stated in the lemma finally follows from an application of Jensen's inequality. $\qquad \square$

# F   PROOF OF PROPOSITION 7

Setting $V = 0$ and plugging in the bound from Proposition 5, we have the following bound from (24) for any round $1 \le t \le T$:

$$\sum_i \mathbb{E}Q_i^2(t) \le 2T + \tilde{O}\Big(\sqrt{N \sum_{t=1}^{T} \sum_i \mathbb{E}Q_i^2(t)} + N^{3/4}T^{5/4}\Big). \tag{37}$$

Define $z_T^2 \equiv \sum_i \sum_{t=1}^{T} \mathbb{E}Q_i^2(t)$. Summing up the inequalities (37) from $t = 1$ to $t = T$, we obtain

$$z_T^2 \le 2T^2 + \tilde{O}(\sqrt{N}Tz_T + N^{3/4}T^{9/4}) \implies z_T = \tilde{O}(N^{3/8}T^{9/8}).$$

Plugging in the above bound in (37), we conclude that

$$\sum_i \mathbb{E}Q_i^2(T) = \tilde{O}(N^{3/4}T^{5/4}) \overset{(\text{Jensen's ineq.})}{\implies} \mathbb{E}Q_i(T) = \tilde{O}(N^{3/8}T^{5/8}), \ \forall i \in [N].$$

# G  PSEUDOCODE FOR THE BANDITQ POLICY IN THE BANDIT FEEDBACK SETUP

As discussed in the main text, the BANDITQ policy in the Bandit feedback setting uses the scale-free MAB algorithm of Putta and Agrawal (2022) in conjunction with the surrogate reward function defined in Eq. (23). The complete pseudocode of the BANDITQ policy is given below in Algorithm 2. In line 12 of the pseudocode, $\text{Breg}_F(x\|y)$ denotes the usual Bregman

---

**Algorithm 2** BANDITQ Policy in the Bandit-feedback setting

1: **Input:** Target reward rate vector $\vec{\lambda}$, $\eta \leftarrow N, \gamma \leftarrow 1/2$, Regularizer $F(q) = \sum_{i=1}^{N}(f(q(i) - f(1/N))$, where $f(x) = -\log(x)$.
2: $\boldsymbol{Q} \leftarrow \boldsymbol{0}, \boldsymbol{p} \leftarrow [1/N, 1/N, \ldots, 1/N], V \leftarrow \sqrt{T}, S \leftarrow 1, \tilde{\boldsymbol{R}} \leftarrow 0.$  ▷ *Initialization*
3: **for each** round $t = 1 : T$: **do**
4:    $\boldsymbol{x} \leftarrow (1-\gamma)\boldsymbol{p} + \gamma/N.$  ▷ *Updating the sampling distribution*
5:    Sample an arm $I_t \in [N]$ from the distribution $\boldsymbol{x}$.
6:    Observe the reward of the selected arm $r_{I_t}(t)$  ▷ *Bandit feedback*
7:    $Q_i = (Q_i + \lambda_i - r_i(t)(I_t = i))^+, \ \forall i \in \mathcal{P}.$  ▷ *Updating the queues*
8:    $r_i' \leftarrow (Q_i + V)r_i(t)\mathbb{1}(I_t = i), \ \forall i$  ▷ *Computing the surrogate rewards*
9:    $\tilde{r}_i \leftarrow \frac{r_i'}{x_i}\mathbb{1}(I_t = i)$  ▷ *Estimating the rewards via the inverse propensity scores (IPS)*
10:   $\tilde{\boldsymbol{R}} \leftarrow \tilde{\boldsymbol{R}} + \tilde{\boldsymbol{r}}$  ▷ *Updating the cumulative estimated surrogate rewards*
11:   $\gamma \leftarrow \min(1/2, \sqrt{N/t}).$
12:   $S \leftarrow S + \eta^{-1} \sup_{q \in \Delta_N}(\langle \tilde{\boldsymbol{r}}, \boldsymbol{q} - \boldsymbol{p}\rangle - \text{Breg}_F(\boldsymbol{q}\|\boldsymbol{p}).$
13:   $\eta \leftarrow N/S$  ▷ *Adaptively choosing the learning rate*
14:   $\boldsymbol{p} \leftarrow \arg\min_{\boldsymbol{q} \in \Delta_N} [F(\boldsymbol{q}) - \eta\langle \boldsymbol{q}, \tilde{\boldsymbol{R}}\rangle]$  ▷ *The FTRL step*
15: **end for each**

---

divergence between the points $x$ and $y$ with respect to the convex function $F(\cdot)$, *i.e.,*

$$\text{Breg}_F(x\|y) = F(x) - F(y) - \langle \nabla F(y), x - y\rangle.$$

# H  EFFICIENT IMPLEMENTATION OF THE OPTIMIZATION MODULE

To speed up the simulation, we implemented a custom-made optimizer for the optimization steps 12 and 14 involved in the BANDITQ algorithm in the bandit-feedback setting. For this, we directly solved the KKT optimality condition, where we computed the optimal KKT multiplier by using the classic Newton-Raphson root-finding algorithm. This empirically resulted in about *two orders* of magnitude speed-up compared to using standard convex optimization packages such as CVX (Grant et al., 2011).

Let $\boldsymbol{r} \in \mathbb{R}^N$ be a given $N$-dimensional real vector. After some simple algebraic manipulations, both the optimization problems in steps 12 and 14 of Algorithm 2 can be expressed in the following form:

$$\text{OPT}(\boldsymbol{r}): \ \max \sum_{i=1}^{N} \log x_i + \langle \boldsymbol{r}, \boldsymbol{x}\rangle \tag{38}$$

Subject to,

$$\sum_i x_i = 1, \ x_i \geq 0, \ \forall i \in [N]. \tag{39}$$

Since the objective function (38) is strictly concave, and the constraint (39) is linear, using the KKT condition, a probability vector $\boldsymbol{x}^*$ is an optimal point for the above problem if and only if there exists a real number $\mu \in \mathbb{R}$ s.t.

$$\frac{1}{x_i^*} + r_i + \mu = 0 \implies x_i^* = -(r_i + \mu)^{-1}, \ \forall i, \tag{40}$$

where $\boldsymbol{x}^*$ satisfies the feasibility condition (39). For the non-negativity constraint on $\boldsymbol{x}^*$, we must have:

$$r_i + \mu < 0 \implies \mu < -\max_i r_i.$$

Finally, we require that

$$\sum_i x_i^* = 1.$$

*i.e.,*

$$\sum_i \frac{1}{r_i + \mu} - 1 = 0. \tag{41}$$

We now use the Newton-Raphson method for solving (41) starting from $\mu^{(0)} = -\max_i r_i - 1$. The algorithm is given below:

---
**Algorithm 3** Custom optimizer for the problem `OPT` $(r)$
---
1: **Input:** $r$, `tolerance` $\leftarrow 10^{-8}$.
2: $\mu \leftarrow -\max_i r_i - 1$, `error` $\leftarrow 1$.
3: **while** `error > tolerance` **do**

$$\mu \leftarrow \mu + \frac{\sum_i \frac{1}{r_i + \mu} - 1}{\sum_i \frac{1}{(r_i + \mu)^2}}.$$

    `error` $\leftarrow |\sum_i \frac{1}{r_i + \mu} - 1|$.
4: **end while**
5: $x_i^* \leftarrow -(r_i + \mu)^{-1}, \; \forall i.$
6: Return $x^*$.

---

# I  ADDITIONAL NUMERICAL RESULTS

## I.1  COMPARISON WITH AN ORACLE POLICY

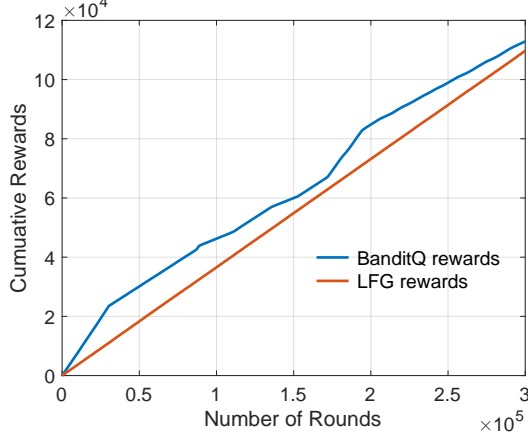

Figure 7: Comparison of reward accrued by the BANDITQ policy and the Oracle LFG policy ($\eta = 100$)

In this section, we compare the performance of the BANDITQ policy with an *Oracle* policy that knows the optimal fraction of pulls of each arm to satisfy the required reward rate constraints. With the given mean reward $\mu$ and the required reward rate vector $\lambda$, the optimal fraction of pulls can be easily computed to be $f_1 = \lambda_1/\mu_1 = 1/2, f_2 = \lambda_2/\mu_2 = 1/3, f_3 = 0, f_4 = 1 - (1/2 + 1/3) = 1/6, f_5 = 0$. In the above computation, we have used the fact that Arm #4 is the most rewarding arm. We emphasize that the oracle policy should have *exact* knowledge of the mean reward vector $\mu$ - a non-zero error in the value of the reward vector either leads to not achieving the target rates or having a linear regret or both.

Note that the online policy proposed by Patil et al. (2021) *cannot* be used with the above profile of fraction of pulls as their policy requires the required fraction of each arm to be at most $1/N-1 = 1/4$. Hence, we use the UCB-based policy proposed

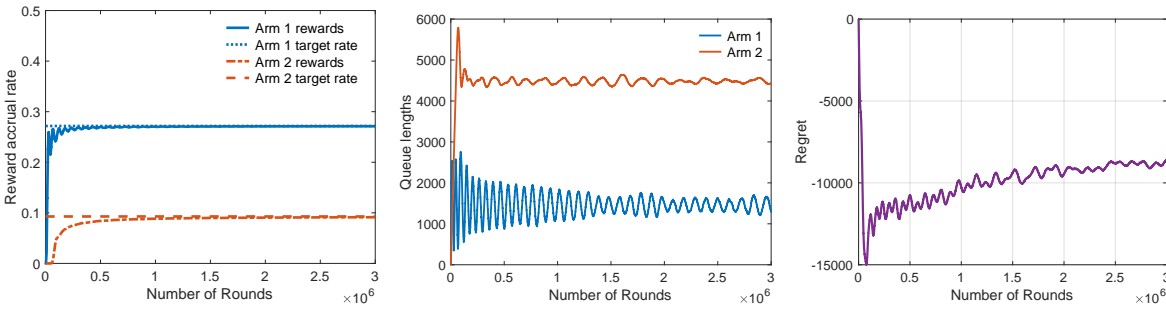

Figure 8: Reward accrual rates in the full-information setting

Figure 9: Queue lengths in the full-information setting

Figure 10: Regret of BANDITQ in the full-information setting

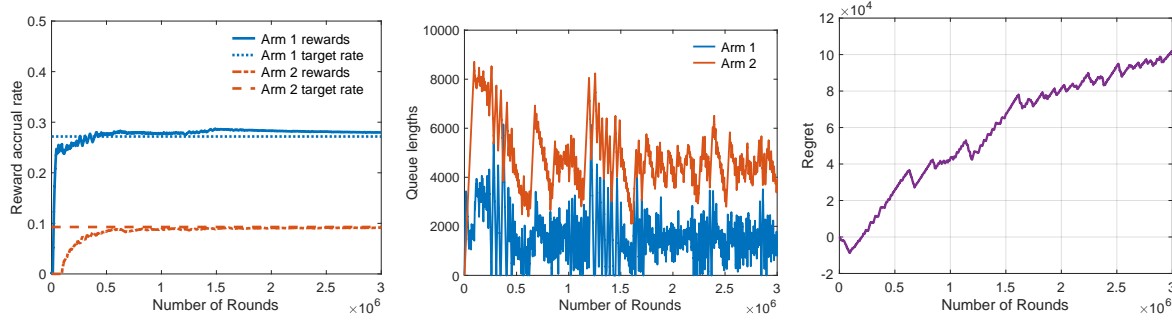

Figure 11: Reward accrual rates in the bandit feedback

Figure 12: Queue lengths in the bandit feedback setting

Figure 13: Regret of BANDITQ in the bandit feedback setting

by Li et al. (2019) called *Learning with Fairness Guarantee* (LFG) as the benchmark. LFG uses queue variables to balance meeting the target fraction of pulls and achieving the small regret. However, as stated in Li et al. (2019, Theorem 2), the best-known regret bound of the LFG policy increases linearly with time.

**Observation:** From Figure 7, we see that the proposed BANDITQ policy yields strictly better cumulative rewards compared to the oracle LFG policy that knows the optimal fraction of arm pulls to meet the given reward rate constraints. This result can be attributed to the fact that the BANDITQ policy directly takes into account the reward realizations through the queue evolutions, whereas the Oracle LFG policy works based only on the expected rewards.

### I.2 LARGE-SCALE SIMULATION WITH $N = 1000$ ARMS

Figures [8-13] show the performance of the BANDITQ policy with $N = 1000$ arms in both full and bandit information settings. The mean rewards $\boldsymbol{\mu}$ for each arm are sampled uniformly at random from the interval $[0, 1]$. As before, we consider two protected arms - arm 1 and arm 2 and set $\lambda_1 = \mu_1/2, \lambda_2 = \mu_2/3$. The plots show that even for a large instance, the BANDITQ policy continues to perform satisfactorily in terms of both regret and achieving the target rates.