# OpenReview forum: "BanditQ:Fair Bandits with Guaranteed Rewards"
_auai.org/UAI/2024/Conference — UAI 2024 poster_

### Official Review · Reviewer_JEvL · 2024-03-18

**Q2-1 Originality-Novelty:** 2
**Q2-2 Correctness-Technical Quality:** 3
**Q2-5 Clarity Of Writing:** 3

**Q1 Summary And Contributions:**

This paper considers a solution for MABs with the constraint that each arm has a pre-specified reward accrual rate. In this work, the authors propose a modified online policy with respect to fairness constraints, called BanditQ, in a full information feedback setting and derive regret bounds. Furthermore, the authors show good empirical performance in simulation experiments.

**Q2-3 Extent To Which Claims Are Supported By Evidence:**

3: Good: the main claims are supported by convincing evidence (in the form of adequate experimental evaluation, proofs, (pseudo-)code, references, assumptions).

**Q2-4 Reproducibility:**

3: Good: key resources (e.g. proofs, code, data) are available and key details (e.g. proofs, experimental setup) are sufficiently well-described for competent researchers to confidently reproduce the main results.

**Q3 Main Strengths:**

The problem that the authors consider (i.e., fairness constraints in MABs) is well-motivated, and the problem setting is novel and the proposed algorithm is convincing.

**Q4 Main Weakness:**

The experimental part seems insufficient because only the case of $N=5$ and $k=2$ is considered. Also, the framing and empirical comparisons in this paper do not adequately characterize/compare against prior works in this space.

**Q5 Detailed Comments To The Authors:**

* There are many alternative notions of fairness mentioned in the related work section, what are the pros and cons of the proposed definition?

* How does BanditQ perform compared to existing works in terms of the proposed performance metric (e.g. the minimum pull rate for each arm)?

**Q9 Complying With Reviewing Instructions:**

Yes

---

> ### Author Rebuttal · Authors · 2024-04-03
>
> $\textbf{On different notions of fairness:}$ As the reviewer pointed out, fairness is indeed a vast topic, with different definitions tailored to particular problems. In the bandit setting, apart from the fairness definition that we provided, which specifies a minimum rate of reward accrual for each arm, other notions of fairness include $\alpha$-fairness (which maximizes the sum of certain non-linear functions of the rewards ), Jain's fairness index, and procedural fairness which specifies the minimum frequency for pulling each arm. Note that Proportional Fairness can be seen as a limiting case of $\alpha$-fairness. The non-linear fairness metrics could be challenging to optimize in the online setting (see the paper [2] in the response above). Please refer to the related works (Section 1.1) for a brief discussion of different fairness notions in connection with bandit problems.
>
> However, to the best of our knowledge, none of the above fairness notions directly guarantee a given minimum cumulative rewards that the arms receive. Since, in the end, it is the cumulative rewards that matter in most applications, ours is the first fair MAB policy that yields a pre-specified minimum reward for each of the arms.
>
>
> $\textbf{On the comparison of BanditQ policy with existing policies:}$ Please note that we indeed numerically compared the performance of BanditQ against the state-of-the-art Learning with Fairness Guarantee (LFG) policy by Li et al. 2019 in terms of the proposed performance metric. Due to a lack of space, this numerical result was included in Appendix L of the accompanying supplementary material. In particular, as Figure 7 in the Appendix shows, empirically, our proposed BanditQ policy offers strictly better cumulative rewards than LFG.
>
> $\textbf{Experimenting with large problem instances:}$ Furthermore, in the following link, we provide a plot of the performance of the BanditQ policy with $N=1000$ number of arms with uniformly distributed mean rewards in [0,1] under the same problem setting (both full and bandit information) as described in the experimental section of the paper (Section 5). Note that we report the result for k=2 only because the resulting plots are cleaner, as here, we need to show the evolution of two queues only. One can also experiment with an arbitrary k with the code provided. Instructions for running the Matlab code are given above.
>
> Link to additional experimental plots: https://drive.google.com/file/d/1pRh--kSOiOAlQ4rmiHSMT2xhdGaR8dUg/view?usp=sharing
>
>
> Link to the code folder (zipped):
> https://drive.google.com/file/d/1qh5TkKaVzeD8-5d2c8HgRXaVtJFY_g1p/view?usp=sharing
>
> Alternative direct link to the code folder
> https://drive.google.com/drive/folders/1ySQGP2-s-81NKoY-1A-khCh1ByBJfiqX?usp=sharing

---

### Official Review · Reviewer_XTEP · 2024-03-22

**Q2-1 Originality-Novelty:** 3
**Q2-2 Correctness-Technical Quality:** 3
**Q2-5 Clarity Of Writing:** 3

**Q1 Summary And Contributions:**

The paper presents algorithms for a bandit framework that maximizes accumulated reward over a given horizon, $T$ while ensuring that each arm receives a certain accumulated reward. For general bandit information setting, they achieve $\mathcal{O}(T^{3/4})$ regret and constraint violation while for full information setting with monotonicity assumption, the bounds improve to $\mathcal{O}(\sqrt{T})$.

**Q2-3 Extent To Which Claims Are Supported By Evidence:**

3: Good: the main claims are supported by convincing evidence (in the form of adequate experimental evaluation, proofs, (pseudo-)code, references, assumptions).

**Q2-4 Reproducibility:**

3: Good: key resources (e.g. proofs, code, data) are available and key details (e.g. proofs, experimental setup) are sufficiently well-described for competent researchers to confidently reproduce the main results.

**Q3 Main Strengths:**

1. The paper is well organized.
2. The main ideas/novelties in the proof are well explained.

**Q4 Main Weakness:**

It is more of a comment rather than pointing out weaknesses.

1. Many papers have recently appeared in the literature on the constrained Markov Decision Process (CMDP) (for example, see [1] and the references therein). The definitions of the regret and constraint violations in the bandit setup seem very similar to those commonly used in the full CMDP setting. It also seems that the constrained bandit framework is a special case of the constrained RL setup. If so, we already have better (at least in terms of $T$) regret and constraint violation bounds in the full CMDP setting (see the comparison table in [1]). In this context, please explain why the CMDP framework cannot treated as the generalization of the constrained bandit framework considered in the paper.

2. Even if the above point turns out to be invalid, comparing your framework (and perhaps the results) with that of the CMDP setting will be helpful.

3. The Lindley recursion (6) looks very similar to the dual update in the CMDP. A qualitative/quantitative comparison will be helpful.

[1] Bai, Q., Mondal, W. U., & Aggarwal, V. (2024). Learning General Parameterized Policies for Infinite Horizon Average Reward Constrained MDPs via Primal-Dual Policy Gradient Algorithm. arXiv preprint arXiv:2402.02042.

**Q5 Detailed Comments To The Authors:**

1. Please see the weaknesses.
2. A discussion on the use of traditional fairness metrics such as Jain's fairness index, proportional fairness, etc. in the bandit literature will be helpful.
3. A discussion and/or comparison to the fair learning in the multi-agent literature will be much appreciated.

**Q9 Complying With Reviewing Instructions:**

Yes

---

> ### Author Rebuttal · Authors · 2024-04-03
>
> $\textbf{1 and 2: Comparing BanditQ with the constrained MDP literature:}$ We thank the reviewer for bringing our attention to the very recent preprint [1], which considers a related problem in the constrained MDP setting. While these two papers consider morally similar problems in different settings, as discussed below, there exist some fundamental differences between them:
>
> 1.	Unlike [1], we consider optimizing over all policies, which need not be continuously differentiable with respect to some parameter in Euclidean space. Hence, in BanditQ, there is no notion of a gradient of the value function, which is fundamental to the policy gradient class of algorithms considered by [1] or the references mentioned therein. Furthermore, we consider multiple constraints, while [1] considers a single long-term constraint.
> 2.	Our analysis is clean in the sense that our main results (Theorem 2 and Theorem 5) hold unconditionally without any non-trivial assumptions. The paper [1] assumes Slater's condition (Assumption 2). This assumption is crucial for their analysis and the validity of the resulting regret bound. However, we do not make any such assumption. Furthermore, unlike Assumption 3 in [1], no assumption on the score function is made in this paper. Furthermore, Assumptions 4 and 5 also do not feature in this paper.
> 3.	Our regret and constraint violation bounds (($O(T^{3/4}$) and $O(T^{3/4}$) respectively, as given in Theorem 5) are stronger than what is claimed in [1] ($O(T^{4/5})$ and $O(T^{4/5})$ respectively).
> 4.	On the technical side, the proof techniques are very different. While the analysis in [1] heavily relies on the stochasticity of the MDP (e.g., to evaluate a policy by taking average over a horizon $H$), our algorithm is constructed primarily based on an adversarial analysis. For example, in the full-information setting, the stochasticity of the problem is used only once – in inequality (a) in Eq (14). Hence, BanditQ can potentially be extended to fully adversarial problems, which cannot be done for policy gradient-type algorithms considered in [1].
>
>
> $\textbf{3. On the relationship between the dual update ([1, Eq. 13]) and Lindley recursion (6):}$ Indeed, these two updates play qualitatively similar roles. To gain some intuition, consider the case where Assumption 2 does not hold, i.e., $\delta=0$. In this case, the dual update equation becomes almost similar to the queue update as the projection operator reduces to the $\max(0,)$ operator. Furthermore, the saddle point optimization ([1, Eq. 14]) is morally similar to the drift-plus-penalty policy used by BanditQ. That being said, it should be emphasized that the algorithms are very different as they solve two distinct problems under different sets of assumptions.
>
> $\textbf{4. On different notions of fairness:}$ Please see our response to the reviewer below.
>
> $\textbf{5. Fair learning in multi-agent bandits:}$ In the complementary multi-agent MAB literature [3], there are multiple agents and a set of $K$ arms. Pulling any of the arms may yield different mean rewards to different agents. Here, the goal is to arrive at a "fair distribution" of arms, which is usually tackled by maximizing some non-linear utility function of the users (e.g., Nash-social welfare [1] or sum of $\alpha$-fairness metric [2]). As discussed, unlike BanditQ, these algorithms do not yield a pre-specified reward to the users. In the revised version, we will include a thorough discussion and comparison of this class of problems.
>
> $\textbf{References}$
>
> [1] Bai, Q., Mondal, W. U., & Aggarwal, V. (2024). Learning General Parameterized Policies for Infinite Horizon Average Reward Constrained MDPs via Primal-Dual Policy Gradient Algorithm. arXiv preprint arXiv:2402.02042.
>
> [2] Sinha, Abhishek, Ativ Joshi, Rajarshi Bhattacharjee, Cameron Musco, and Mohammad Hajiesmaili. "No-regret Algorithms for Fair Resource Allocation." Advances in Neural Information Processing Systems 36 (2024).
>
> [3] Hossain, Safwan, Evi Micha, and Nisarg Shah. "Fair algorithms for multi-agent multi-armed bandits." Advances in Neural Information Processing Systems 34 (2021): 24005-24017.

---

### Official Review · Reviewer_JMWJ · 2024-03-23

**Q2-1 Originality-Novelty:** 3
**Q2-2 Correctness-Technical Quality:** 3
**Q2-5 Clarity Of Writing:** 3

**Q10 Ethical Concerns:**

No ethical concerns.

**Q1 Summary And Contributions:**

This paper considers a fair prediction problem in MAB with a guaranteed minimum rate of accrual of rewards for certain arm set. Different from the previous settings, the constraint considered in this paper is a minimum rate of reward accruals for the arms. The author provide BANDITQ algorithm, providing regret upper bounds for both full-information and bandit feedback settings, and achieves nearly optimal regret when having monotonicity assumption.

**Q2-3 Extent To Which Claims Are Supported By Evidence:**

3: Good: the main claims are supported by convincing evidence (in the form of adequate experimental evaluation, proofs, (pseudo-)code, references, assumptions).

**Q2-4 Reproducibility:**

3: Good: key resources (e.g. proofs, code, data) are available and key details (e.g. proofs, experimental setup) are sufficiently well-described for competent researchers to confidently reproduce the main results.

**Q3 Main Strengths:**

The new settings introduced in paper may have some practical applications. It presents an algorithm with theoretical guarantees on regret and rate violation. The main idea is using the non-negative state variable $Q_i$'s to guide the algorithm towards fairness while use standard updates for the sample distributions. Additionally, empirical evaluations are provided to support the proposed approach.

**Q4 Main Weakness:**

1. The regret upper bound presented in Theorem 2 seems to be loose. If the order of $Regret_t = O(t^{3/4})$ in Theorem 2 is indeed tight, then the summation of regrets $\sum_{t=1}^{T} Regret_t$ should be of the order $O(t^{7/4})$, leading to an average regret of $O(t^{3/4})$, which than the $O(\sqrt{T})$ shown in Proposition 2. As the average regret is a reweighted version of the regret with larger weights assigned to the initial rounds, comparing these two bounds shows that the algorithm performs better at initial than after collecting enough samples, which is not true.

2. There is a lack of a lower bound analysis. The paper compares its regret upper bound with the minimax regret bound for the unconstrained problem in the order of $T$ only, with a substantial gap between them. And the lower bound for rate violation is absent.

**Q5 Detailed Comments To The Authors:**

1. There are some typos in the paper:
- (3) the summation should range from $i=1$ to $N$
- (5) $\mathbb{P}$ should be $\mathcal{P}$
- The special case in Theorem 2. Setting $V=\sqrt{T}$ should end with $\text{Regret}_T=O(\max\{T^\{3/4\}, \sqrt\{NT\}\})$.
- The special case in Theorem 5. Setting $V=\sqrt{T}$ should end with $\text{Regret}_T=O(\max{N^{3/4}T^{3/4}, N\sqrt{T}})$.
- Theorem 3 should be state in terms of $T$

2. The definition of the regret in (3): The regret defined in (3) can be negative (as shown in Figure 3), can we define it as supremum over all possible $x^*$ with all analysis remains valid.

3. The reason why it is difficult to adapt well-known bandit policies to this problem is not well stated. For instance, consider a simple algorithm that maintains a set $A$ containing under-explored arms (those not meeting the fairness constraint plus a small constant). At each round, if $A$ is non-empty, the algorithm randomly selects an arm from $A$; otherwise, it chooses an action according to the Upper Confidence Bound (UCB). Why does this seemingly straightforward algorithm fail in this setting?

4. Definition of $Q_i(t)$ in Equation (6): While $Q_i(t)$ is defined as non-negative, which makes sense, but what will happen if we define $Q_i(t)= (\sum_{t=1}^{T} (\lambda_i -r_i(t)x_i(t)))^{+}$, it seems to perform better as it is more accurately accessing the rate violation.

5. Proposition 3 and Proposition 4 seem to be not meaningful, since setting $V=0$ makes regret bound to be infinite.

6. The monotonicity assumption in Assumption 1 seems related to the specific algorithm. Is there evidence or reasoning to suggest that the BanditQ algorithm satisfies this assumption? Figures 2 and 5 seem to violate this assumption.

**Q9 Complying With Reviewing Instructions:**

Yes

---

> ### Author Rebuttal · Authors · 2024-04-03
>
> $\textbf{On the tightness of the bounds:}$ Please note that in this problem, we want to control the regret and rate violations simultaneously, and in general, these two objectives are at odds with each other. If one only cares about regret as in the classical problem, by setting the queue lengths to zero, it can be easily shown that BanditQ yields the usual $O(\sqrt{T})$ regret bound.
>
> In fact, our main motivation for stating Proposition 2 was to show that BanditQ indeed performs better in terms of time-averaged regret. However, Proposition 2 does not immediately imply that the regret bound in Theorem 2 can indeed be improved.
>
> $\textbf{On the lower bound:}$ We agree with the reviewer on this point. However, to establish a minimax lower bound for this problem, we need to prove a joint (two-dimensional) lower bound that holds for the regret and the constraint violations simultaneously. Currently, we are not aware of any such lower bound in the literature. In this paper, we are mainly concerned with the achievability part of the problem.
>
> 1.	$\textbf{On the typos:}$ We thank the reviewer for carefully pointing out the typos.
>
> (a)	(3) We use the standard summation convention where missing upper and lower limits of a summation means that the summation is taken over all terms.
>
> (b)	Theorem 3 holds for any $t\leq T$. We will clearly mention this in the statement of the theorem.
>
>
> We will fix the other typos in the revised version.
>
>
> 2.	$\textbf{Definition of regret:}$ We are afraid that even in the standard unconstrained bandit setting, where $x^*$ is taken to be the best-fixed action in hindsight, the regret can be negative. This is because, while the online policy can change its action on every round by adapting to the reward vectors, the benchmark is forced to use the same action on every round. Hence, in the case of favourable reward vectors, the algorithm can perform better than the offline static benchmark.
>
> Secondly, allowing $x^*$ to be an arbitrary probability distribution in Eqn. (3) would not work because, in this problem, we want to achieve sublinear bounds for both the violation and regret. Hence, we need to restrict the benchmark action $x^*$ to the set of all probability distributions, which satisfies the rate constraints for each arm in expectation. Otherwise, proving a sublinear bound for both regret and constraint violation would be impossible.
>
> 3.	$\textbf{Adapting UCB-like policies:}$  Vanilla UCB-like policies ensure that a fraction of pulls of all non-optimal arms converge to zero. However, in this problem, we want to maintain pre-specified constant reward rates for all arms.  Coming back to the interesting scheme suggested by the reviewer, we believe a simple uniform sampling of arms in A would probably not work as one also needs to ensure a low regret. Hence, the exploration bonus and sampling probabilities must be carefully designed to arrive at a competitive performance bound. In fact, BanditQ does something similar as it simultaneously weighs in the current rate violation (via Q's) and rewards (weighted by $V$) before pulling arms on each round. We leave exploring the UCB-class of policies for this problem as a future research topic.
>
> 4.	$\textbf{Alternative definition to Q(t):}$ Note that, as Eqn (4) defines, we want to bound the rate violation over any consecutive interval within the horizon of length $T$. Likewise, our rate violation bound holds for any subinterval in $[T]$.
>
> To point out a potential issue with the proposed alternative definition $Q_i(t)= (\sum_{t=1}^T (\lambda_i – r_i(t)x_i(t)))^+$, imagine a policy which satisfies the constraints by a large margin in the first half of the horizon. Hence, although $Q_i(T/2)$ would be zero, the inner summation would be roughly $-O(T)$. Now, if the algorithm stops serving the ith arm in the second half (i.e., $x_i(t)=0, \forall t> T/2$), $Q_i(t)$ would still continue to be zero for a long time in the second half due to the massive negative summation accumulated in the first half. Defining $Q$ variables via Lindley recursion (6) immediately eliminates this issue.
>
> 5.	$\textbf{Proposition 3 and Proposition 4}$ pertain to the problem when we only need to satisfy the target reward accrual rates for each of the arms while not worrying about maximizing the cumulative reward. This is a non-trivial and meaningful problem on its own.
>
> 6.	$\textbf{On the Monotonicity assumption:}$ To give an example, assume that the policy satisfies each constraint with equality on each round, \emph{i.e.}, $\lambda_i = \mu_i \mathbb{E} x_i(t),  \forall i$. Since $\max(0,.)$ is convex, using Jensen’s inequality, we have $\mathbb{E}Q_i(t) = \mathbb{E}(Q_i(t-1)+\lambda_i-r_i(t)x_i(t))^+ \geq  (\mathbb{E}Q_i(t-1)+0)^+ = \mathbb{E}Q_i(t-1), ~ \forall i \in [N].$
>
> Note that even if the monotonicity assumption is not satisfied, the $O(T^{\frac{3}{4}})$ regret bound from Theorem 2 continues to hold.

---

### Official Review · Reviewer_Lpmf · 2024-03-23

**Q2-1 Originality-Novelty:** 3
**Q2-2 Correctness-Technical Quality:** 3
**Q2-5 Clarity Of Writing:** 3

**Q1 Summary And Contributions:**

In this paper, the authors study the problem of a stochastic multi-armed bandit with an additional fairness constraint. An algorithm based on the BanditQ is proposed. The authors provided a solid theoretical analysis for the proposed algorithm which proves that the algorithm achieves a regret and a non-asymptotic metric rate violation penalty of $O(T^{3/4})$. Furthermore, the authors also provide theoretical results in the full-information setting.

**Q2-3 Extent To Which Claims Are Supported By Evidence:**

3: Good: the main claims are supported by convincing evidence (in the form of adequate experimental evaluation, proofs, (pseudo-)code, references, assumptions).

**Q2-4 Reproducibility:**

2: Fair: key resources (e.g. proofs, code, data) are unavailable but key details (e.g. proof sketches, experimental setup) are sufficiently well-described for an expert to confidently reproduce the main results.

**Q3 Main Strengths:**

* The problem of multi-armed bandits under the fairness constraint, which the paper investigates, is an important problem with broad potential applications and is thus within the scope of this conference in general.

*  The paper is fairly well-written and clear. The algorithms are clean and natural.

**Q4 Main Weakness:**

* The paper lacks thorough comparisons with previous works regarding theoretical bounds. While some comparisons are provided in the related work section, it is challenging to figure out comparable results under identical settings.

* While the algorithm appears to draw inspiration from prior works, the underlying intuition behind its utilization remains inadequately explained.

* Experimentally, the algorithms are evaluated exclusively on a small instance with $N=5$ arms, which diminishes practicality. Consequently, doubts arise regarding the scalability of implementing the proposed algorithm on larger instances.

**Q5 Detailed Comments To The Authors:**

Solving the multi-armed bandit problem with fairness constraints is a very important topic and there are many applications in the real world. The paper is clearly written and well-motivated and I only have two concerns of this paper: (1). Firstly, the algorithm's underlying intuition remains ambiguous. Maybe the authors can explain more clearly the intuition behind the design of state variables potentially through the elucidation of illustrative toy examples. (2). It seems that this paper shares a similar idea as the one by Neely. While it is argued that the new technique introduced in this paper is to adapt the asymptotic stochastic setting of Neely (2010) to the non-asymptotic adversarial setup, the extent of novelty and technical contribution of this new technique remains ambiguous. Elucidating the conceptual intuition and the primary challenges encountered in achieving this adaptation could enhance the paper's clarity and contribution.

**Q9 Complying With Reviewing Instructions:**

Yes

---

> ### Author Rebuttal · Authors · 2024-04-03
>
> $\textbf{Reproducibility:}$ Please note that complete proofs of all theoretical claims are available in the Appendix. We also release a Matlab implementation of the experiments in the link given below.
>
> $\textbf{Intuition:}$ The queueing evolution (6), which denotes the backlog dynamics in a queue for a given arrival and departure process, is used to capture the cumulative rate violation. Note that, in this problem, we want to simultaneously control both the regret and violation metrics, which result in a non-trivial two-dimensional online objective. We transform this requirement to a scalar objective by constructing a suitable potential function, which, in turn, requires us to control a linear combination of the cumulative violations and rewards (Line 7 in Alg 1). Here, the parameter $V$ denotes a trade-off between the objectives. Finally, we use an off-the-shelf online learning subroutine to control the scalar metric. The non-trivial part is to show that the above procedure works.
>
> $\textbf{Comparison with related work:}$
>
> We are not aware of any previous work that considers an identical or comparable setting to ours in the bandit feedback setting, which is the main focus of this paper. In the full-information setting, while there exists a body of work that guarantees both regret and constraint violation (Neely and Yu 2017, Yu et al. 2017), none of them reduces the problem to a black-box online learning subroutine as we do.  As discussed in the related work section, they propose problem-specific policies upon additionally assuming Slater's condition.
>
> $\textbf{Performance of BanditQ for Large N:}$
>
> We emphasize that the proposed algorithm is highly scalable as its runtime scales linearly with $N$. We reported a small instance only because the optimal fraction of pulls for each arm can be calculated by hand. However, this in no way limits scaling our algorithm to large instances.
>
> A moderately large instance with $N=1000$ arms with random mean rewards can be simulated for $T=$ 3 million rounds under three minutes on a standard laptop in both full-information and bandit settings. We provide a working Matlab script in the link below. We also include performance plots for different random instances of the problem in the same folder. We encourage the reviewer to try a few different parameter settings to confirm the scalability of BanditQ.
>
> $\textbf{Instruction to run the code:}$ Please download the entire code folder from the link provided below. To experiment with the full information setting, please run the script "BanditQ_full_info.m". To experiment with the bandit information setting, please run the script "BanditQ_bandit_expt.m". The parameters $N,k,\mu$ can be changed at the beginning of the script. The plots will be saved as pdf files in the same folder.
>
> Link to additional experimental plots:
>
> https://drive.google.com/file/d/1pRh--kSOiOAlQ4rmiHSMT2xhdGaR8dUg/view?usp=sharing
>
> Link to the code folder:
> https://drive.google.com/file/d/1qh5TkKaVzeD8-5d2c8HgRXaVtJFY_g1p/view?usp=sharing
>
> Alternative link:
> https://drive.google.com/drive/folders/1ySQGP2-s-81NKoY-1A-khCh1ByBJfiqX?usp=sharing
>
>
> $\textbf{Comparison with Neely, 2010:}$
>
> There are several differences in the problem settings considered by the stochastic network utility maximization framework of Neely 2010 and this paper.
>
> 1.	$\textbf{Different nature of the bounds:}$ Neely 2010 gives asymptotic queue length bounds which, for large $T$, scales as $O(1/\epsilon)$, where $\epsilon$ denotes the "gap" between the arrival rate vector to the capacity region of the network (see Fig 3.1 (b) in Neely 2010 where this concept is illustrated). Hence, if this gap is zero, the queue-length bound diverges. This divergence is expected intuitively (also from Little's law) as any system that operates close to the capacity region should have a very large number of backlogs. All queue length bounds given in the Neely 2010 are of this form.
>
> On the other hand, in our regret and rate violation bounds, this gap does not feature at all – our bounds remain finite for all finite $T$, even when condition (2) (which is comparable to a capacity bound) holds with equality. Needless to say, one needs an entirely different kind of analytical and algorithmic technique, vastly distinct from Neely's Greedy Max-weight policy, that yields the above bounds. Although we use a quadratic Lyapunov function argument, we achieve these refinements by supplementing it with additional techniques from adversarial online learning.
>
>
> 2.	$\textbf{Bandit information:}$ The policy proposed by Neely 2010 works in the full-information setting. It is a greedy policy, known as Max-weight for single-hop networks or Backpressure in multi-hop networks, which opportunistically serves the most backlogged queues, assuming that all arrivals and departures up to any time are known. Hence, these policies do not involve any exploration part, which is essential to our problem in the bandit information setting.

---

### Official Review · Reviewer_Aj5W · 2024-03-25

**Q2-1 Originality-Novelty:** 3
**Q2-2 Correctness-Technical Quality:** 3
**Q2-5 Clarity Of Writing:** 4

**Q1 Summary And Contributions:**

The paper tackles the relevant fairness problem of appropriately sampling the arms to ensure that the underperforming arms are able to get a minimum reward rate. Both the full information setting as well as bandit settings are considered. Experiments show that the rates are indeed achieved with the proposed algorithm BanditQ.

Overall: The paper is well-written and all the main ideas, theorems, results are present in the main body. The problem is highly relevant and the results are novel.

**Q2-3 Extent To Which Claims Are Supported By Evidence:**

4: Excellent: all claims are supported by very convincing evidence (in the form of comprehensive experimental evaluation, rigorous mathematical proofs, detailed (pseudo-)code, precise references, well-motivated and realistic assumptions) and the authors deliver what they promise.

**Q2-4 Reproducibility:**

4: Excellent: key resources (e.g. proofs, code, data) are available and key details (e.g. proof sketches, experimental setup) are comprehensively described for competent researchers to confidently and easily reproduce the main results.

**Q3 Main Strengths:**

(i) The theory supports both full-information and bandit settings and obtains regret bounds which seem reasonable. The reward rate is explicitly kept track of through queueing theory and thus enables them to manage the fairness constraints.

(ii) Their framework can reuse any MAB algorithms as long as they satisfy certain properties.

(iii) Experiments are extensive and consider both full-information and bandit settings. Additional experiment with an oracle policy is also provided showing the benefit of the approach.

**Q4 Main Weakness:**

(a) There was no explicit dependence on the number of fair arms in the bound (K, the size of set P).
(b) Lower bounds are not provided or discussed as far as I can tell.

**Q5 Detailed Comments To The Authors:**

(1) How do the regret bounds change with the hardness of the fairness constraints in terms of either the rates and/or the number of arm constraints?
(2) How does BanditQ work on adversarial problems in terms of experiments?

**Q9 Complying With Reviewing Instructions:**

Yes

---

> ### Author Rebuttal · Authors · 2024-04-03
>
> $\textbf{Dependence of the bounds on the number of arms:}$ Even in the unconstrained bandit setting $(K=0)$, the minimax regret is lower bounded by $\sqrt{NT}$. Thus, the dependence of the regret bound on $N$ is unavoidable. Hence, for simplicity, we decided to state our bounds in terms of a single parameter N by trivially upper-bounding $|P|=K$ by $N$. Thus, there is no dependence of the reported bounds on $K$.
>
> In view of this, Theorem 2 gives an explicit dependence of the regret on the number of arms $(N)$ – it is $O(\sqrt{N}T^{3/4})$. Although Theorem 2 does not mention the variation of the violation bound with N, it can be easily read off from Eqn (22) from the derivation of Theorem 2. The violation bound can be computed to be $O(N^{1/4}T^{3/4})$. In the Bandit case, the statement of Theorem 5 and Proposition 4 gives dependence on both N and T for both regret and violation.
>
> $\textbf{Lower bounds:}$ After the statement of Proposition 1, we mentioned that in the full information case, a minimax regret lower bound (without any constraints) is known to be $O(\sqrt{T\log N})$ (Zhao and Chen 2019; Theorem 4). Proposition 2 shows that we match this bound with respect to $T$ in terms of average regret, and Theorem 3 shows that this bound is matched by the terminal regret under Assumption 1.
>
> Minimax regret lower bound $O(\sqrt{NT})$ is well-known for unconstrained bandit problems. Unfortunately, we are not aware of any non-trivial joint lower bound for regret and constraint violations in either the full-information or bandit setting. This is indeed an interesting question that we will leave as a future research direction.
>
> $\textbf{Variation of the regret bound with respect to the hardness of the problem: }$
>
> One aspect of the hardness of the problem can be characterized by the minimum gap between the required reward rate and the mean rate of any arm. In this paper, we provide instance-independent bounds which do not depend on this gap at all. Hence, as long as this gap is non-negative (however small), the regret and constraint violation bounds continue to hold. In particular, the bounds do not blow up as this gap approaches zero! Finally, we comment that this aspect of hardness can be characterized by instance-dependent bounds which we do not study in this paper.
>
> $\textbf{Adversarial rewards: }$
>
> We have carried out some preliminary experiments with real-world wireless datasets where the target rates for the arms are chosen in such a way that they can be satisfied in the long run in hindsight. The presence of wireless fading makes the rewards non-stationary, and they can be approximated as adversarial. The performance of BanditQ for this dataset has so far turned out to be satisfactory in the sense that it is able to satisfy the rate constraints while providing substantial total rewards (throughput in this case).

---

### Meta-Review · Area_Chair_spuM · 2024-04-21

This paper makes a clear cut improvement over interactive learning setting in both full information and bandit feedback models. The authors also responded in a satisfactory way to the reviewers.